# Effect of high dust amount on surface temperature during the Last Glacial Maximum: A modelling study using MIROC-ESM

Rumi Ohgaito[1], Ayako Abe-Ouchi[2,1], Ryouta O'ishi[2], Toshihiko Takemura[3], Akinori Ito[1], Tomohiro Hajima[1], Shingo Watanabe[1], Michio Kawamiya[1]

[1]Japan Agency for Marine-Earth Science and Technology, Yokohama, 236-0001, Japan
[2]Atmosphere Ocean Research Institute, University of Tokyo, Kashiwa, Chiba 277-8564, Japan
[3]Research Institute for Applied Mechanics, Kyushu University, Fukuoka, 816-8580 Japan

*Correspondence to*: Rumi Ohgaito (ohgaito@jamstec.go.jp)

**Abstract.** The effect of aerosols is one of many uncertain factors in projections of future climate. However, the behaviour of mineral dust aerosols (dust) can be investigated within the context of past climate change. The Last Glacial Maximum (LGM) is known to have had enhanced dust deposition in comparison with the present, especially over polar regions. Using the Model for Interdisciplinary Research on Climate Earth System Model (MIROC-ESM), we conducted a standard LGM experiment following the protocol of the Paleoclimate Modelling Intercomparison Project phase 3 and sensitivity experiments. We imposed glaciogenic dust on the standard LGM experiment and we investigated the impacts of glaciogenic dust and of non-glaciogenic dust on the LGM climate. Global mean radiative perturbations by glaciogenic and non-glaciogenic dust were both negative, consistent with previous studies. However, glaciogenic dust behaved differently in specific regions, e.g., it resulted in less cooling over the polar regions. One of the major reasons for reduced cooling is the ageing of snow or ice, which results in albedo reduction via high dust deposition, especially near sources of high glaciogenic dust emission. Although the net radiative perturbations in the lee of high glaciogenic dust provenances are negative, warming by ageing of snow overcomes this radiative perturbation in the Northern Hemisphere. In contrast, the radiative perturbation due to high dust loading in the troposphere acts to warm the surface in areas surrounding Antarctica, primarily via the longwave aerosol–cloud interaction of dust and it is likely the result of the greenhouse effect attributable to the enhanced cloud fraction in the upper troposphere. Although our analysis focused mainly on the results of experiments using the atmospheric part of the MIROC-ESM, we also conducted full MIROC-ESM experiments for an initial examination of the effect of glaciogenic dust on the oceanic general circulation module. A long-term trend of enhanced warming was observed in the Northern Hemisphere with increased

glaciogenic dust; however, the level of warming around Antarctica remained almost unchanged, even after extended coupling with the ocean.

**1 Introduction**

The Last Glacial Maximum (c.a. 21,000 years before present; LGM), which is the most recent period featuring maximum expansion of the land ice sheets in the Northern Hemisphere, has been investigated thoroughly using various paleo-proxy records and via modelling studies (Braconnot et al., 2007a,b, Kageyama et al., 2006, 2017). Climate modelling is an essential tool in investigations seeking to clarify the mechanisms of climate change, as stated in the Intergovernmental Panel on Climate Change (IPCC) assessment reports (IPCC, 2013). Therefore, it is especially important to evaluate the capability of numerical models to capture past climatic conditions.

Paleo-proxy data and modelling studies are both required for proper understanding of past climates; however, the focus of this study is on modelling. General circulation models (GCMs) are one of the tools used most widely for investigation of the mechanisms of both climate and climate change. The improvement of computational resources has allowed the development of models with high complexity that permit interactive coupling of various climatic components. In comparison with proxy data, previous modelling experiments targeting the LGM have tended to underestimate the magnitude of cooling, especially over high latitudes (Masson-Delmotte et al., 2006, 2010). The importance of feedback related to dust and vegetation has been identified in Chapter 5 of the IPCC's Fifth Assessment Report (IPCC, 2013).

It is recognized that uncertainty over the effect of aerosols is one of the most important factors regarding the radiative perturbation in estimates of global warming. Mineral dust is the most abundant atmospheric aerosol, even in the present climate. For example, Mahowald et al. (2010) investigated the trend of the amount of atmospheric dust in the 20[th] century based on observations and

modelling. They reported correlation between an increase of desert dust and a net negative radiative perturbation. Examination of proxy data has suggested clear enhancement of dust during the LGM, which was especially pronounced at high latitudes, i.e. reaching levels more than 20 times greater than the present day over Antarctica (Lambert et al., 2008, Lamy et al., 2014, Dome Fuji Ice Core Project members 2017). Although the enhancement of dust deposition was found less over lower latitudes, it was still several times

5 higher in comparison with the present day (Winckler et al., 2008).

Although earlier studies (Mahowald et al., 1999, Lunt and Valdes, 2002, Claquin et al., 2003) have estimated higher dust amounts during the LGM in comparison with the pre-industrial (PI) period, dust amounts over Antarctica have tended to be underestimated. Claquin et al. (2003) estimated the radiative perturbation at the top of the atmosphere (TOA). They reported a cooling effect attributable to dust, but they also found a warming effect due to dust deposition on snow. Later, Mahowald et al. (2006a,b) estimated

the glaciogenic dust flux and the aerosol–radiation interaction. Their standard LGM experiment simulated underestimation of dust deposition flux, especially over high latitudes, in comparison with the DIRTMAP proxy data archive (Kohfeld and Harrison 2001). Then, they considered the effect of sources of glaciogenic dust surrounding the ice sheets and glaciers. Such areas are defined to generate substantial amounts of glacial flour during glacial periods (Bullard et al. 2016). The study considered the emission of various fluxes of dust from these glaciogenic source areas and a best fit to the DIRTMAP deposition distribution was obtained. Although this

estimate could conceal other possible and non-introduced processes of dust sources, it constitutes an important step forward in the determination of a reasonable representation of both the atmospheric loading and the depositional distribution of dust during the LGM. However, they did not estimate the effects of aerosol–cloud interaction. Takemura et al. (2009) used the Model for Interdisciplinary Research on Climate (MIROC) Atmospheric GCM (AGCM) with an online aerosol module to determine both the aerosol–radiation and the aerosol–cloud interactions for LGM and PI periods at both the surface and the tropopause. However, they underestimated the

amount of dust deposition over Antarctica, probably because they did not consider glaciogenic dust. Lambert et al. (2013) used two General Circulation Models coupled with online aerosol models and obtained underestimated dust flux and radiative forcing. This underestimation was global, but more pronounced over the polar regions and they suggested the possibility that it contributes to an underestimation of polar amplification for LGM and future projections. Yue et al. (2011) used an AGCM to estimate the aerosol–radiation interaction for dust and

they reported an evident cooling effect. Albani et al. (2014) supposed high erodibility areas to obtain better representation of LGM dust. They also highlighted the importance of the optical properties and size distribution of dust aerosols. In comparison with the control setting, Sagoo and Strelvmo (2017) applied an emission factor of 3.4 to the dust emissions in an LGM level $CO_2$ experiment (i.e., the land sea mask and ice sheets were unchanged from the control) to mimic the high dust situation during the LGM and they estimated the aerosol–cloud interaction. Hopcroft et al. (2015) investigated the aerosol–radiation interaction at the TOA using an

AGCM and the land module of an earth system model (ESM), based on which they suggested the necessity of further analyses of aerosol–cloud interaction as future work. They also summarized the global mean dust emissions and loadings of the PI and LGM periods reported in previous studies. It was suggested that the amount of dust is highly dependent on the model used, not only for the LGM but also for the PI. The latest review of previous studies is also in Albani et al. (2018).

Another aspect of dust is related to ageing of the snow surface, which possibly modulates the surface temperature via albedo reduction.

Krinner et al. (2006) discussed the importance of the ageing effect of snow, particularly over eastern Siberia. Their ageing scheme was based on that of Warren and Wiscombe (1981) and Wiscombe and Warren (1981). Moreover, Ganopolski et al. (2010) simulated the glacial–interglacial cycle using an intermediate complexity model, in which the ageing effect was implemented via simple scaling. Previous studies have not included dynamic ocean in this context, so the impacts on global ocean circulation are unknown.

In summary, we claim that the evaluation of the total effect of dust on the LGM surface temperature is incomplete. Therefore, this study addresses the problem by incorporating the effects of aerosol–radiation interaction, aerosol–cloud interaction, snow ageing, and dust–ocean interaction. We undertook AGCM simulations and full ESM simulations of the LGM with sensitivity experiments targeting the effects of dust on climate.

The following section explains the modelling and experimental set-ups. The resulting estimations of dust amount and dust depositional distribution are presented in Sect. 3.1 and the influence of dust on surface temperature is described in Sect. 3.2. To investigate how dust might modulate the atmospheric state, the radiative perturbation attributable to dust is described in Sect. 3.3 and the effect of glaciogenic dust on the ocean is discussed in Sect. 3.4. The results of the simulations are summarized and discussed in Sect. 4.

## 2 Model and experimental design

### 2.1 Description of the MIROC-ESM

The MIROC-ESM (Watanabe et al., 2011) used in this study was the version submitted to both the Coupled Model Intercomparison Project phase 5 (CMIP5) and the Paleoclimate Modelling Intercomparison Project phase 3 (PMIP3). The resolution of the atmosphere in the model is T42 with 80 vertical levels, while that of the ocean is about $1°$ ($256 \times 192$). Although the model is capable of computing the amount of $CO_2$ in the atmosphere, we prescribed the level of atmospheric $CO_2$ in our experimental set-up. The spatially explicit individual-based Dynamic Global Vegetation Model (SEIB-DGVM) (Sato et al., 2007) was implemented to simulate global vegetation dynamics and terrestrial carbon cycling in the system, but it returns only the leaf area index (LAI) to the Minimal Advanced Treatments of Surface Interaction and Runoff (MATSIRO) land module (Takata et al., 2003). In this model, the SEIB-DGVM received several variables from the AGCM, but it returned only the carbon flux to the atmosphere. Also implemented was the Spectral Radiation–

Transport Model for Aerosol Species (SPRINTARS) on-line aerosol module (Takemura et al., 2000, 2002, 2005, and 2009), which explicitly treats organic, black carbon, and mineral dust, sea-salt aerosols, and sulfate and its precursor gases. This module was coupled with the radiation and cloud microphysical schemes to calculate the aerosol–radiation and aerosol–cloud interactions. In the calculation of the former, refractive indices depending on wavelengths, size distributions, and hygroscopic growth were considered. The refractive

index of dust aerosols was taken from Deepak and Gerber (1983) but its imaginary part was reduced for consistency with recent measurements of weaker shortwave absorption to $1.530\text{-}2.00 \times 10^{-3}$i at 0.55 micro-meter dust (Takemura et al. 2005). Number concentrations of both cloud droplets and ice crystals are prognostic variables, as are their mass mixing ratios, and the changes in their radii and precipitation rates were calculated. Thus, the aerosol–cloud interaction was taken into account (See Takemura et al. (2009) for more details). The processes controlling dust generation are the surface wind, vegetation type, soil moisture, LAI, and snow cover.

Once dust is generated, it is transported via the atmospheric circulation and deposited via the processes of wet/dry deposition and gravitational settling. In this study, glaciogenic dust was imposed for the sensitivity experiments. The generation of glaciogenic dust flux followed the estimate of Mahowald et al. (2006a). This flux was added as time-invariant sources into the simulations and are not dependent on modelled land surface or atmospheric conditions.

In the MATSIRO module, the effect of dirt in snow (i.e., snow ageing) was considered based on the work of both Yang et al. (1997)

and Warren and Wiscombe (1981). The magnitude of dirt concentration at the snow surface was varied to fit an observed relation between snow albedo and dirt concentration (Aoki et al., 2006). The dirt concentration in snow was calculated from the deposition fluxes of dust and soot calculated in the SPRINTARS module. The relative strength of the absorption coefficients for dust and soot were weighted as a function of the deposition fluxes to obtain radiatively effective amounts of dirt in the snow.

## 2.2 Experimental design

We performed eight experiments: five using the AGCM part of the MIROC-ESM and three using the full MIROC-ESM. The specific experiments labelled PI.a and PI.e represent the 1850 A.D. control climate of the PI era, with PI.e having been submitted to CMIP5. The previous 100-year climatology of sea surface temperature (SST) and of sea ice of the period submitted to CMIP5 was used as boundary conditions for PI.a. The experiments labelled LGM.e and LGM.a represent the LGM climate following the PMIP3 protocol (Abe-Ouchi et al., 2015). The LGM.e experiment was submitted to CMIP5/PMIP3 (Sueyoshi et al., 2013). The LGM.a experiment was the AGCM experiment using the SST and sea ice taken from the PMIP3 LGM experiment (LGM.e). The LGM.e experiment was extended for a further 800 years beyond the PMIP3 period (Fig. 1). The LGMglac.a experiment was a new experiment based on the same conditions as LGM.a, but with an additional glaciogenic dust flux following Mahowald et al. (2006a). The LGMglac.naging.a and LGM.naging.a experiments had the same settings as LGMglac.a and LGM.a, but without the effect of snow ageing. The LGMglac.e experiment was the full ESM version of LGMglac.a, which branched from the LGM.e experiment 40 years prior to the period submitted to CMIP5/PMIP3 (Fig. 1). The glaciogenic dust flux from each area was set identical to the estimates of Mahowald et al. (2006a) and the emission areas were defined as shown in supplementary Fig. A to follow their work as closely as possible, i.e. the three areas of strongest emission were the Pampas of South America, central North America, and eastern Siberia. In contrast to non-glaciogenic dust, the emission of glaciogenic dust was independent of dust emission conditions and it was emitted constantly for consistency with the dust flux in Mahowald et al. (2006a) (Table 1(b)). Once emitted into the atmosphere, the treatment of glaciogenic dust was identical to non-glaciogenic dust. The integration of LGMglac.e was performed for 940 years. Table 2 lists the details of all the experiments.

## 3 Results

### 3.1 Dust amount and comparison with data archives

The emission flux of dust (g m$^{-2}$ y$^{-1}$) is shown in Fig. 2 for the PI.a, LGM.a, and LGMglac.a experiments. For the PI.a experiment, the major dust sources are the Saharan, Arabian, Gobi, and Taklamakan deserts. A minor source is also found in the mid-latitude region of South America. While these dust sources look reasonable based on the present-day situation, there is too little dust emission from the other plausible dust sources such as Australia, southern Africa, and southwestern North America. The wet bias over these areas in the PI.a experiment leads to excess vegetation, which prevents dust emission, and persists in the LGM.a and LGMglac.a experiments. In the LGM.a and LGMglac.a experiments, the dust emission flux in the Saharan, Gobi, and Taklamakan deserts is significantly enhanced, which is the result of a windier and drier climate during the LGM, with additional emission flux evident from northern Siberia. In contrast, the emission flux from South America is reduced, which is probably because of increased soil moisture resulting from enhanced precipitation in this region. For the LGMglac.a experiment, glaciogenic dust emission is evident surrounding the extended ice sheets during the LGM. The total emission amount is 2540 (Tg y$^{-1}$) for the PI.a experiment, 7250 (Tg y$^{-1}$) for the LGM.a experiment, and 13,400 (Tg y$^{-1}$) for the LGMglac.a experiment. The total simulated emissions and atmospheric loads are listed in Table 1.

The global dust budget can be compared with the findings of previous studies. Hopcroft et al. (2015) summarized it in their Table 1. They clarified that the dust amount is highly dependent on the model, not only for the LGM experiments but also for the PI experiments. Our emission and load values fall in the middle of the ranges determined by previous studies. However, they are close to those of Takemura et al. (2009) for PI.a and LGM.a, probably because the models adopted are from the same model family and use the same

aerosol module. The emission of LGMglac.a is close to that of Mahowald et al. (2006a), most likely because we adopted their glaciogenic dust, but the load of LGMglac.a (39 Tg) is about 60 % of Mahowald's loading (62 Tg), which suggests overestimation of immediate dust deposition rates near the source areas (Fig. 4) attributable to our assumption of the independence of dust emission from wind speed. The change in the zonal mean dust loading in the atmosphere for the ratios LGM.a/PI.a and LGMglac.a/PI.a is

shown in Fig. 3(a) and 3(b), respectively. In the LGM.a experiment, the dust mass concentration in the Northern Hemisphere is enhanced, but decreased in the Southern Hemisphere compared with the PI.a experiment. In contrast, the mass concentration is enhanced significantly in both the northern and the southern high latitudes in the LGMglac.a experiment. The glaciogenic dust reached higher levels of the troposphere in the Southern Hemisphere compared with the Northern Hemisphere. This can be attributed to the different conditions of the strong dust sources. In the Southern Hemisphere, they are exposed to stronger winds because of the lack of

continental land, whereas in the Northern Hemisphere, the strong sources of glaciogenic dust are located over continents that are subject to lower wind speeds. The distribution of dust deposition for each experiment is shown in Fig. 4(a)–(c) and the ratio to PI.a is shown in Fig. 5 for comparison with the archives of ice and sediment core data, as indicated by the coloured circles (Kohfeld et al., 2013, Albani et al., 2014). The scatter plots shown in Fig. 4(d)–(f) compare the data with the modelled deposition rate at the grids corresponding to the data locations. The colours and mark types are used for categorization according to the area and the type of core

data. Reasonable correlation is seen for the PI.a experiment, except in the grids over the Southern Ocean, which are mostly located in the southern Pacific Ocean region. The main source of the dust deposited in this region is expected to be Australia (Li et al., 2010, Albani et al., 2012), where our model underestimates the emission. In the LGM.a experiment, the dust deposition flux is underestimated in North America, Eurasia, the South Pacific, the Southern Ocean, and Antarctica. In contrast, in the LGMglac.a experiment, the underestimation is generally improved. The model–data linear correlation coefficients in the logarithmic scale are

0.79, 0.62, and 0.80 for the PI.a, LGM.a, and LGMglac.a experiments, respectively. The differences in the deposition flux between the PI.a and PI.e experiments, LGM.a and LGM.e experiments, and LGMglac.a and LGMglac.e experiments are almost negligible.

## 3.2 Surface temperature at LGM and the effect of glaciogenic dust

The surface temperature anomaly for LGM.a-PI.a is presented in Fig. 6 (a). The cooling is about 2 ℃ over the tropics and increase towards high latitudes. The most pronounced cooling is seen over the ice sheets in the northern hemisphere. This general view is also seen in Fig. 6 (b) which shows temperature anomaly for LGMglac.a to PI.a. Fig. 7 (b) and (c) show the anomaly of the downward radiation for LGM to PI for these experiments. Cooled atmosphere at LGM results in reduced longwave reaching the earth's surface is consistent with the distribution of the temperature anomalies in Fig. 6.

Now, we focus on imposed glaciogenic dust. The surface temperature at the height of 2 m is influenced by glaciogenic dust and the difference of LGMglac.a relative to LGM.a is presented in Fig. 6 (c). The warming (i.e., less cooling compared with the PI.a results) is pronounced in the high latitudes in contrast to the expectation of the likely cooling effect of the dust (IPCC, 2013).

The changes in the LGMglac.a result relative to the LGM.a result for the net, longwave, and shortwave downward radiation at the surface are presented in Fig. 7 (a), (d) and (g). The figures represent the total effect of the atmospheric loading of glaciogenic dust on radiation toward the earth surface. Figure 7(g) shows a negative anomaly in shortwave radiation near the strong sources of glaciogenic dust, as well as in the northern high latitudes and the edge of Antarctica. In contrast, a positive anomaly of longwave radiation in the LGMglac.a experiment is pronounced around Antarctica and in the northern high latitudes (Fig. 7 (d)). While the negative anomaly in shortwave radiation dominates the net change near the areas of glaciogenic dust emission, the positive longwave anomaly dominates the region surrounding Antarctica. The radiative perturbation attributable to the glaciogenic dust is detailed in the next section.

Figure 8 shows that warming of LGMglac.a−LGM.a south of 55° S is evident without the inclusion of the effects of the ageing of snow (LGMglac.naging.a−LGM.naging.a). This suggests the warming around Antarctica is not the result of snow ageing but that it follows from the change in the radiation balance in the atmosphere. Moreover, the magnitude of the warming is not significantly affected by ocean coupling (LGMglac.e−LGM.e). In contrast, more than 80 % of the warming in the Northern Hemisphere is the result of ageing of the snow surface, as is evident by inspection of the LGMglac.naging.a−LGM.naging.a results (Fig. 8). The high dust deposition rate reduces the surface albedo as shown in Supplementary Fig. B and leads to reduction of reflected shortwave radiation, which overcomes the cooling effect of the dust loading in the atmosphere, resulting in warming (Fig. 6 (c)). The warming in the Northern Hemisphere is most pronounced over eastern Siberia and central North America, where large amounts of glaciogenic dust are deposited, and therefore where the albedo of the LGMglac.a experiment is reduced significantly. The snow in the LGMglac.a experiment thaws earlier in the year than in the LGM.a experiment over eastern Siberia. Substantial snowmelt over a large area within this region accelerates warming via albedo reduction. This is consistent with the results of Krinner about snow ageing preventing the accumulation of snow in this region. In contrast, in central North America, the snow is reduced compared with the LGM.a experiment but it is still significantly higher than the PI.a experiment. The position of the −2 °C isotherm averaged over June–August, which is the threshold of ice sheet retreat–extension (Ohmura et al. 1996), shifted northward by about 1° latitude, which is significantly less than the model resolution. Therefore, the effect of our dust flux on climate is lesser melting of the Laurentide Ice Sheet. However, we question whether the model is able to represent the appropriate ageing of snow under such a high dust deposition flux. As this is beyond the scope of this study, further evaluation of the effects of snow ageing are required.

### 3.3 dust-related aerosol-radiation and aerosol-cloud interactions

The aerosol–radiation and aerosol–cloud interactions were estimated using the same method as Takemura et al. (2009). The aerosol–radiation interaction was estimated based on the difference between a standard experiment including dust impacts and another experiment under the same conditions but without the dust affecting radiation. The aerosol–cloud interaction was estimated based on the difference between a standard experiment and another experiment under the same condition but without dust at all.

The net global mean radiative perturbation (aerosol–radiation and aerosol–cloud) of dust is one of cooling at the earth's surface for all the experiments, i.e., PI.a: $-0.99$ W m$^{-2}$, LGM.a: $-1.50$ W m$^{-2}$, and LGMglac.a: $-1.71$ W m$^{-2}$. The breakdown of the LGM experiments relative to the PI experiment for the change in the global mean radiative perturbation is listed in Table 3. The net change of the global mean aerosol–radiation interaction at the TOA is slightly positive for the LGM.a−PI.a and it amounts to 0.12 W m$^{-2}$ for the LGMglac.a−PI.a results. Albani et al. (2018) summarized the result from previous studies about aerosol-radiation interaction at TOA. Our positive anomaly at TOA is located around the upper end of the previous studies ranging from about -3 to 0.1 W m-2. On the other hand, the change at the surface is negative both with ($-0.21$ W m$^{-2}$) and without ($-0.30$ W m$^{-2}$) glaciogenic dust. The change at the surface is of similar magnitude to the findings of previous studies (e.g., $-0.25$ and $-0.56$ W m$^{-2}$ with and without glaciogenic dust in Mahowald et al. (2006b), $-0.23$ W m$^{-2}$ in Takemura et al. (2009), and $-0.26$ W m$^{-2}$ in Albani et al. (2014)), and it is caused primarily by changes in shortwave radiation. The net change of the global mean aerosol–cloud interaction at the TOA for the LGM.a−PI.a result is $-0.36$ W m$^{-2}$. Both the shortwave and the longwave radiation increased with glaciogenic dust, resulting in a net change of $-0.39$ W m$^{-2}$. At the surface, without glaciogenic dust, there is net negative reduction in comparison with the TOA. With the inclusion of glaciogenic dust, however, the change at the surface is slightly more negative than the change at the TOA. Considering

the total effect of dust, but without glaciogenic dust, the radiative perturbation change at the TOA relative to the surface is small, whereas the inclusion of glaciogenic dust results in surface cooling via aerosol–radiation interaction.

Figure 9 shows the spatial distribution of radiative perturbation by dust at the TOA, which has a smaller difference between the LGMglac.a and LGM.a results compared with the surface (Fig. 10 (a)). At the TOA, although the influence of glaciogenic dust from the Pampas region is distributed over the Southern Ocean, the positive longwave and negative shortwave radiation almost cancel each other out. There are local negative effects over the strong sources of glaciogenic dust but the amplitudes are much smaller than at the surface (Figs. 9 (a) and 10 (a)). Supplementary Fig. C shows the LGMglac.a–LGM.a anomaly of aerosol–radiation and aerosol–cloud interactions for the TOA and the surface; it also presents the same information but without the snow ageing effect. The panels clarify that the effect of snow ageing is independent from radiative perturbation by dust load in the atmosphere. The figure also clarifies that the anomaly of the aerosol–radiation interaction tends to be significant at the level of 0.1 W m$^{-2}$, whereas the significance of the anomaly of the aerosol–cloud interaction is difficult to determine. Nevertheless, the positive anomaly around Antarctica at the surface is significant. Therefore, although glaciogenic dust changes the TOA radiation budget only marginally, it heats/cools the atmosphere and causes a greater change in the radiation budget at the surface. The global mean change resulting from the addition of glaciogenic dust is cooling ($-0.19$ W m$^{-2}$), but with local atmospheric heating over the high latitudes. Hereafter, we investigate the changes in the spatial distribution and strength of radiation at the surface under different climatic conditions.

Figure 10 shows the change of the net radiative perturbation due to dust at the surface for the LGMglac.a−LGM.a, LGMglac.a−PI.a, and LGM.a−PI.a experiments. The aerosol–radiation interaction dominates near the massive dust sources, e.g., the Sahara Desert. Except for such regions, the aerosol–cloud interaction dominates the radiative perturbation. The addition of glaciogenic dust acts to reduce shortwave radiation. The negative radiative perturbation is distinct near the emission areas. In contrast, for longwave radiation,

a general positive radiative perturbation resulting from glaciogenic dust is obvious, especially near the strong sources of dust and at the edge of Antarctica. The negative shortwave radiation forcing overcomes the positive longwave radiation forcing near the sources of glaciogenic dust. However, the positive longwave radiative perturbation plays a role in the regions surrounding Antarctica. The higher dust loading in the higher troposphere in the Southern Hemisphere promotes the generation of cloud ice nucleation and high-

level clouds, especially in the regions surrounding Antarctica, likely resulting in an enhanced greenhouse effect, which warms the lower troposphere (Figs. 3(c) and 11). Because the dust deposition flux of the standard LGM.a experiment is higher than the PI.a experiment in the Northern Hemisphere but lower in the Southern Hemisphere, the impact of glaciogenic dust might be more efficient in the Southern Hemisphere. Sagoo and Strelvmo (2017) reported global mean cooling in a "high" dust experiment, consistent with our results (Table 3). The discrepancies could arise because of different cloud ice nuclei schemes, of their experimental setting (no

change of land from their control)  and because their sources of high dust emission were located mainly in desert areas, whereas our glaciogenic dust sources are located in the high latitudes.

**3.4 Influence of glaciogenic dust on the ocean**

We extended the LGM.e experiment by 800 years beyond the original PMIP3 period (Fig. 1) and the LGMglac.e experiment was conducted for 940 years. Because the temperatures become quasi-stable after year 600 in Fig. 1, the average of the final 300 years is

used for the analyses. The strength of the Atlantic Meridional Overturning Circulation (AMOC) of LGM.e reduced by about 10 Sv in the analysis period compared with the spin-up period and LGMglac.e. The strength of the abyssal cells (Supplementary Fig. D) is more stable but with differences of a few Sverdrup between LGM.e and LGMglac.e reflecting the AMOC state. The surface air temperature and SST changes according to the LGMglac.e–LGM.e results are presented in Fig. 12. The zonal mean anomaly of air temperature

over land and scatter plots of the anomaly of the proxy data (Bartlein et al., 2011) and of the anomaly of the corresponding model grids are shown in Supplementary Fig. E. It illustrates the level of agreement between the model and the proxy archives. Pronounced discrepancy is evident in the northern high latitudes around 70 °N with some proxy data over Alaska suggesting warmer temperatures than PI, which is not resolved in all our LGM experiment and the other LGM experiment in PMIP3 models. Although the differences between LGM.e and LGMglac.e appear minor in comparison with the pollen proxy archive, LGMglac.e generally exhibits slightly closer agreement with the proxy data.

Warming of the SST by the increased air temperature for LGMglac.e compard to LGM.e is obvious in the northern high latitudes, but the magnitude of the SST change is mostly below 0.5 °C. Locally strong warming along the Gulf Stream can be attributed to differences in the strength of the thermohaline circulation. Although investigation of the effect of dust on the thermohaline circulation is left for future work, we note there might be a possibility of an effect of strong snow ageing in the Northern Hemisphere. In contrast, almost no change is calculated in the SST around Antarctica (Fig. 12 (f)), which confirms that warming around Antarctica is not attributable to a change in the temperature of the ocean surface. Even after the extended integration times of our simulations, the high plateau over the Antarctica, which is often the location of ice core sites, does not warm further (e.g., see circled letters in Fig. 12(a)–(c)). The LGMglac.e cooling from the PI.e results for this area is largely within the range of observational estimates (−7 to −10 °C) (Stenni et al., 2010, Uemura et al., 2012).

The SST anomaly in both the LGM.e-PI.e and the LGMglac.e-PI.e experiments appear reasonable in comparison with the LGM SST reconstruction shown by coloured circles (MARGO project members, 2009) (Fig. 12 (d) and (e)). Local cooling of the ocean temperature is seen in the lee of the source of glaciogenic dust in Argentina, which would be caused by the negative radiative perturbation (Figs. 7 and 10(a)).

The zonal mean potential temperature and salinity anomalies in the Atlantic and Pacific oceans for the LGM.e-PI.e and LGMglac.e-PI.e experiments are presented in Supplementary Figs. F and G. The positive anomalies in the Northern Hemisphere in Supplementary Figs. F(c) and G(c) are attributable to the difference in the strength of the AMOC between LGM.e and LGMglac.e. The minor negative anomaly in the upper 100 m around 30° S in the Atlantic basin can be attributed to the effect of glaciogenic dust from the Pampas area.

## 4 Conclusions and discussion

This study used the MIROC-ESM to investigate the effect of mineral dust aerosols on the glacial climate. The representations of climatology by the PI.a and PI.e simulations are considered reasonable for a state-of-the-art ESM (Watanabe et al. 2011). The cooling evident in the LGM.e experiment in comparison with the PI.e results is also generally comparable with paleo-proxy archives (Fig. 12). The net radiative effect of global mean dust during the LGM is negative, which is the same trend as reported in previous studies (Mahowald et al. 2006b, Albani et al. 2014, Hopcroft 2015, Sagoo and Strelvmo 2017). The global mean value is dominated by high emission of dust from subtropical deserts. Takemura et al. (2009) suggested an LGM-PI anomaly of $-0.9$ W m$^{-2}$ for the global mean aerosol–cloud interaction, whereas our anomaly is $-0.36$ W m$^{-2}$ (Table 3), even though the results are based on models from the same model family. This difference in the global mean value is derived mainly from the different boundary conditions used for the PI experiment. The SST used by Takemura et al. (2009) (Ohgaito et al. 2009; Fig. 1) over the warm pool was about 1° warmer than the SST used in this study (Sueyoshi et al. 2013; Fig. 4), suggesting different convective activity and consequently, different amounts of

cloud ice and cloud water. This tropical difference influences the global mean value, suggesting that the SST bias of the control experiment could affect both regional and global mean radiative perturbations.

The focus of this study was on the high latitudes, with investigation of the effect of glaciogenic dust based on new LGMglac.a and LGM.a experiments using the AGCM part of the MIROC-ESM. The effect of the addition of glaciogenic dust on climate is evident mainly as warming in the high latitudes. The effect of mineral dust aerosol on climate is highly uncertain but cooling is relatively likely (IPCC, 2013). Our results suggest the effect of dust on climate is dependent on background condition. However, our glaciogenic dust worked different from that demonstrated by Mahowald et al. (2006b) in the zonal mean. Especially for the northern high latitudes, areas are warmed via albedo reduction because of snow ageing and because of prolonged disappearance of snow at certain periods, which is especially pronounced in eastern Siberia. Although the longwave radiative perturbation is negative near the strong sources of glaciogenic dust flux, the snow ageing effect overcomes this cooling, resulting in a net increase in temperature. The possibility of overestimation of ageing of snow effect or our simple emission method may influence the result.

The warming effect resulting from the addition of glaciogenic dust is also seen in areas surrounding Antarctica; however, it is not attributable to snow ageing but to longwave aerosol–cloud interactions. Accounting for this effect would alter the distribution of the scatter evident in Fig. 5.5(d) in the IPCC's Fifth Assessment Report, which shows the correlation of eastern Antarctic cooling during the LGM with the future projection.

We adopted additional dust sources from Mahowald et al. (2006a, b) as a first step, where their glaciogenic dust flux was identified as a best fit to the DIRTMAP data archive. Nevertheless, as noted, their deposition flux does not correspond well to new proxy data at locations in the Southern Ocean. However, in our case, this mismatch can also be attributed to a feature of our model, i.e., insufficient dust emission from Australia and South Africa, which is caused mainly by overestimation of soil moisture and the resulting excess of

vegetation. It should be noted that there is still a possibility of contamination by ice rafted debris at the edge of sea ice extent. Our study draws attention to the high dust loading over the Southern Ocean that affects the increase in surface temperature in areas surrounding Antarctica, implying the necessity of investigation of climate sensitivity to the amount of dust emission in future work. However, over the Southern Ocean, SST is affected minimally (Fig. 8) by the surface radiation change (Figs. 7(a) and 10(a)), probably

because of the large heat capacity of the ocean.

Glaciogenic dust was imposed constantly in this study, which is not realistic. In reality, temporal variability of glaciogenic dust should be dependent on changes both in wind speed and in the threshold wind friction velocity at which dust emission is initiated. Thus, the independence of dust emission from wind speed might cause overestimation of dust deposition rates at the grids close to emission areas and under low atmospheric loading. However, our results are in good agreement with the measurements of deposition flux in

general. It will be necessary to implement a better scheme for glaciogenic dust in subsequent research. Sagoo and Strevmo (2017) prescribed a globally "idealized high" dust emission factor for their LGM-like experiment. Because our glaciogenic dust sources are located in the high latitudes, the influence of glaciogenic dust emission on the surface temperature around Antarctica is likely more pronounced in our simulation results.

In the tropics, the effect of enhanced dust input on the surface temperature is similar to what Mahowald et al. (2010) reported in their

study of the mid- to late 20[th] century but with contrasting effects at high latitudes. The major difference is that dust is enhanced at low latitudes, i.e., the Sahara–Sahel drought in the 20[th] century perturbation compared with the additional high dust inputs at high latitudes in our study, where the background albedo is high because of the extended areas of snow and ice cover.

In the MIROC-ESM, snow cover in the PI.e (PI.a) experiment tends to persist in boreal spring over Siberia in comparison with reanalysis data (Supplementary Fig. H). This positive bias might influence the change we see in the LGM.e (LGM.a) and LGMglac.e (LGMglac.a) experiments.

The strong effect of snow ageing is especially significant in the Northern Hemisphere. Because snow ageing has been tuned to fit modern observations in Hokkaido, Japan (Aoki et al., 2003, 2006) in the MIROC-ESM, a strong dust provenance near snow-covered areas is lacking, e.g., as in the glaciogenic dust situation seen in eastern Siberia. Therefore, evaluation of the quantitative influence of snow ageing using various observational sites is needed. The albedo impurity relationship provided by Aoki et al. (2003, 2006), in which ageing starts to work when the impurity is ≥10 ppmw, explains the reason for the considerable snow ageing in the Northern Hemisphere but lack of snow ageing over Antarctica. The deposition flux over Antarctica is 3–4 orders of magnitude smaller than the regions of high dust emission in the Northern Hemisphere. The threshold of activation of snow ageing is in between the high dust deposition in the Northern Hemisphere and the low deposition flux around Antarctica.

Although we were unable to treat the effect of Fe supply to the ocean in this model, activating the Fe-fertilization effect and enhancing the amount of plankton would influence $CO_2$ uptake, especially over the Southern Ocean (Martin, 1990). Improved representation of the distribution of dust deposition is possible as a boundary condition for off-line biogeochemical models to investigate $CO_2$ uptake, e.g., in a more realistic version of the experiments by Oka et al. (2011). Further investigation of the non-negligible effect of the change in the size distribution of dust as identified by Albani et al. (2014), Mahowald et al. (2014), and Hopcroft et al. (2015) might also be necessary.

Plant functional types are considered in the dynamic vegetation module but not returned to the land module in the MIROC-ESM; i.e., the climate–vegetation interaction is limited. The importance of full vegetation coupling was highlighted by O'ishi and Abe-Ouchi

(2013), who suggested the necessity for future models to evaluate the changes of plant functional types and especially, their effect on dust cycles.

Under global warming, the amount of dust emission remains uncertain (Woodward et al., 2005, Tegen et al., 2004, Jacobson and Streets, 2009, Liao et al., 2009, Mahowald et al., 2006a, Ito and Kok, 2017). Therefore, improving the understanding of dust processes

5 in models of the past climate would be a practical way to reduce the uncertainty of projections into the future.

**Author Contributions:** ROH, AA and ROI discussed the paleoclimate motivations on LGM dust and experimental design. ROH designed, conducted, analysed experiments and documented the manuscript. TT and AI advised the analyses on the aerosol module. SW, TH, and MK developed MIROC-ESM. All authors contributed discussions.

**Acknowledgements:**

This research was supported by the "Integrated Research Program for Advancing Climate Models (TOUGOU program)" from the Ministry of Education, Culture, Sports, Science and Technology (MEXT), Japan, and was also partly supported by the Japan Society for the Promotion of

15 Science (JSPS) KAKENHI under Grant Number 17H06104 and JP17H06323. The model experiments were conducted on the Earth Simulator of JAMSTEC. The authors are grateful for the help and inspiring discussions offered by the MIROC development team of JAMSTEC/U, Tokyo/NIES, especially Dr. K. Takata on her help to understand the MATSIRO land module. We appreciate two anonymous reviewers for their constructive and shrewd comments which improved this study significantly. We also thank James Buxton MSc from Edanz Group (www.edanzediting.com/ac) for English reviewing of the manuscript.

**Conflicts of Interest:** The authors declare that they have no conflict of interest.

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

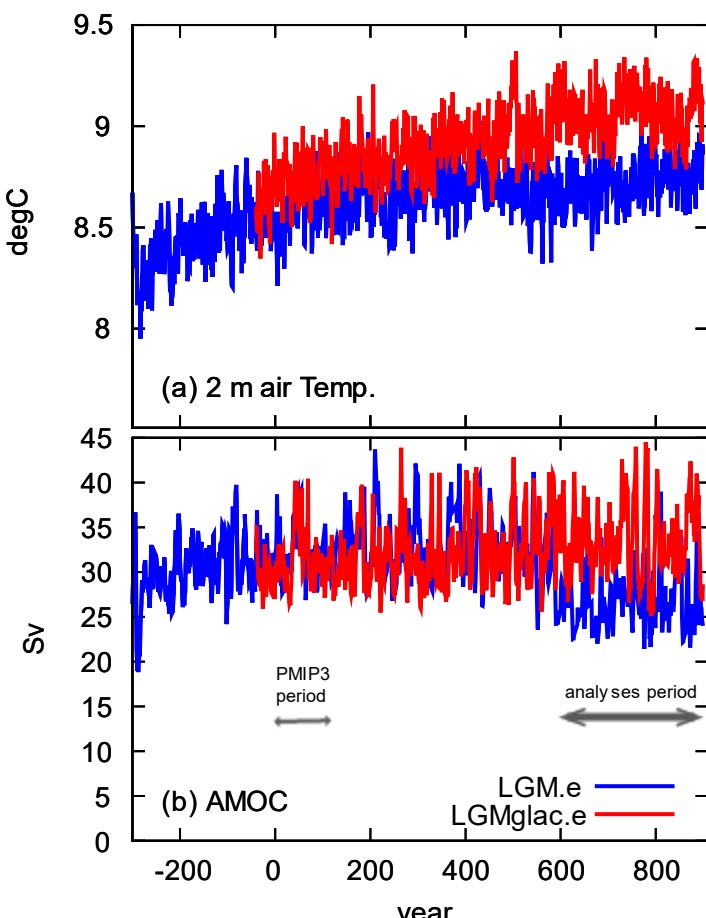

**Figure 1: Time series of (a) global mean annual mean temperature at 2 m height (°C) and (b) peak strength of the Atlantic meridional overturning circulation (AMOC; Sv) for LGM.e and LGMglac.e. The year zero was set to the beginning of the period submitted to CMIP5.**

## dust emission flux (g/m²/y)

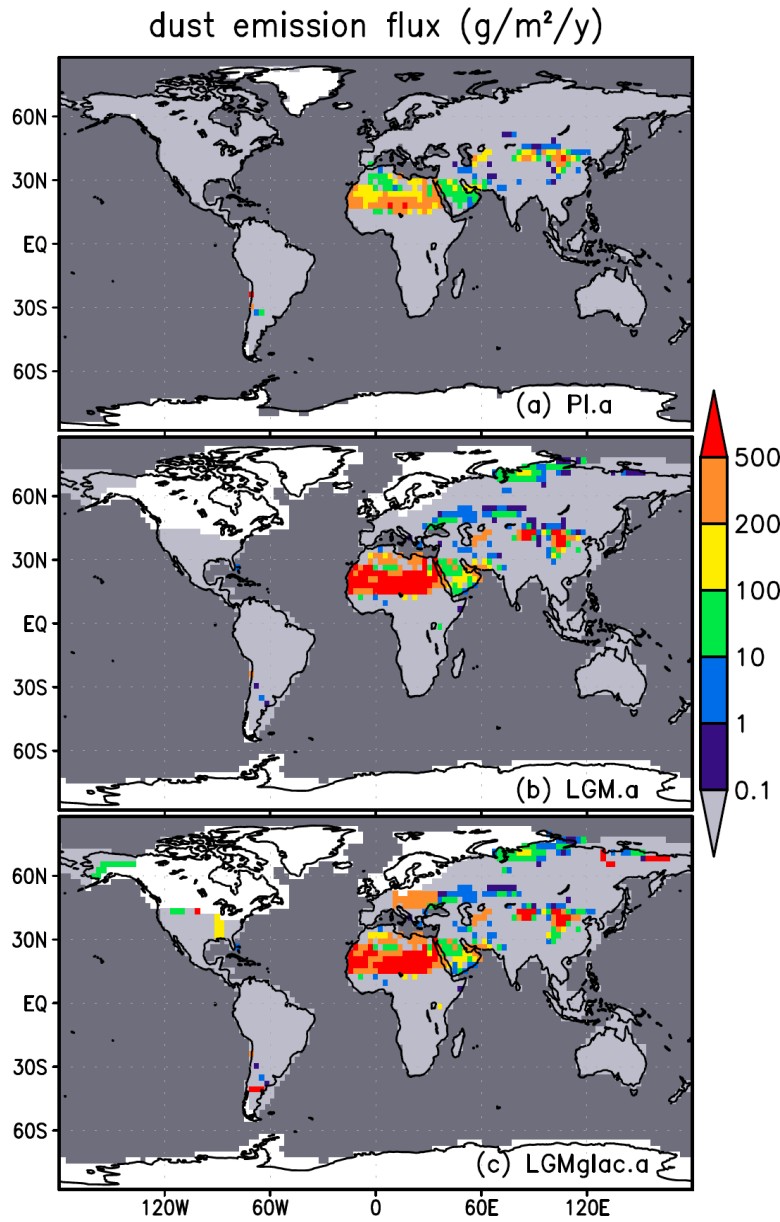

**Figure 2: Dust emission flux (g m$^{-2}$ y$^{-1}$) for (a) PI.a, (b) LGM.a, and (c) LGMglac.a. Ocean areas are dark grey and ice sheets are white.**

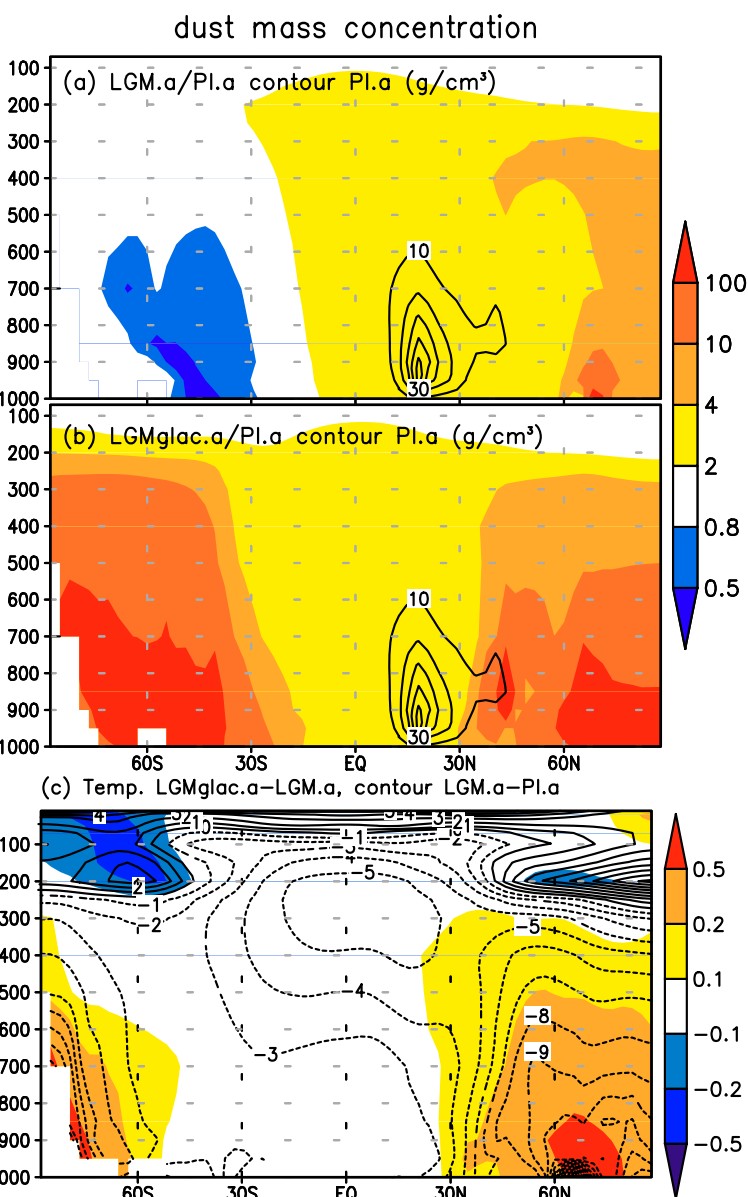

**Figure 3: All panels are zonal mean height plots. Ratio of the dust mass concentration for (a) LGM.a/PI.a, (b) LGMglac.a/PI.a, and (c) temperature change for LGMglac.a–LGM.a. Contour lines in (a) and (b) show the dust mass concentration for PI.a (g cm$^{-3}$) and in (c) the temperature change for LGM.a–PI.a (oC).**

# deposition flux (g/m²/y)

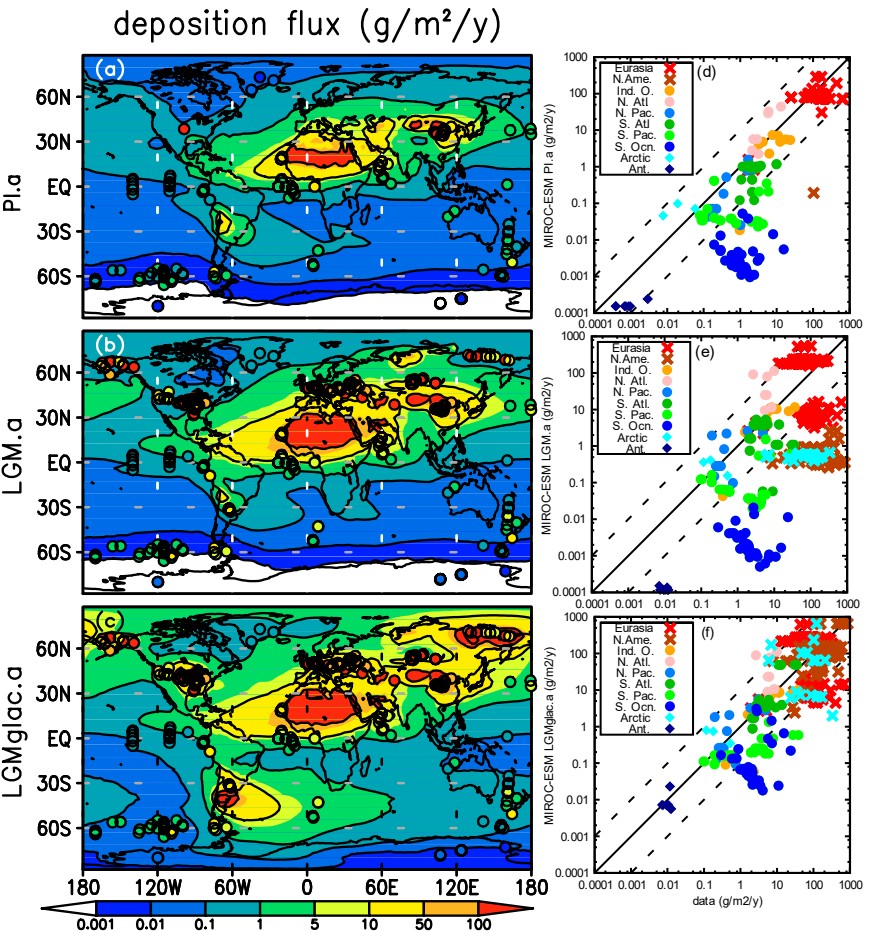

**Figure 4: Model–data comparison of dust deposition flux (g m⁻² y⁻¹) estimated from the ice and sediment core data archives obtained from Kohfeld et al. (2013) and bulk data from Albani et al. (2014): (a) PI.a, (b) LGM.a, and (c) LGMglac.a. Model–data scatter plots for (d) PI.a, (e) LGM.a, and (f) LGMglac.a. Colours and marks represent areas and core types, i.e., red: Eurasia, brown: North America, orange: Indian Ocean, pink and light blue: Atlantic and Pacific oceans in the Northern Hemisphere, respectively, green and light green: Atlantic and Pacific oceans in the Southern Hemisphere, respectively, blue: Southern Ocean, turquoise blue: Arctic, and dark blue: Antarctica. Crosses, circles, and diamonds represent terrestrial, marine core, and ice core sediments, respectively.**

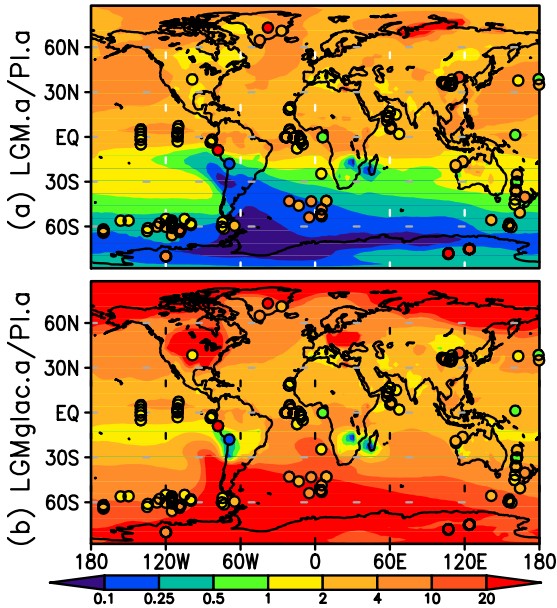

**Figure 5: Model–data comparison of ratio of dust deposition flux estimated from the ice and sediment core data archives obtained from Kohfeld et al. (2013) and Albani et al. (2014): (a) LGM.a/PI.a and (b) LGMglac.a/PI.a.**

# air temperature at 2 m
## (a) LGM.a−PI.a (°C)

## (b) LGMglac.a−PI.a (°C)

## (c) LGMglac.a−LGM.a (°C)

**Figure 6: Difference of surface temperature at 2 m height (degree C) for (a) LGM.a-PI.a (b) LGMglac.a-PI.a and (c) LGMglac.a–LGM.a. Change is considered not significant at the 95 % confidence level in the hatched area based on a t-test.**

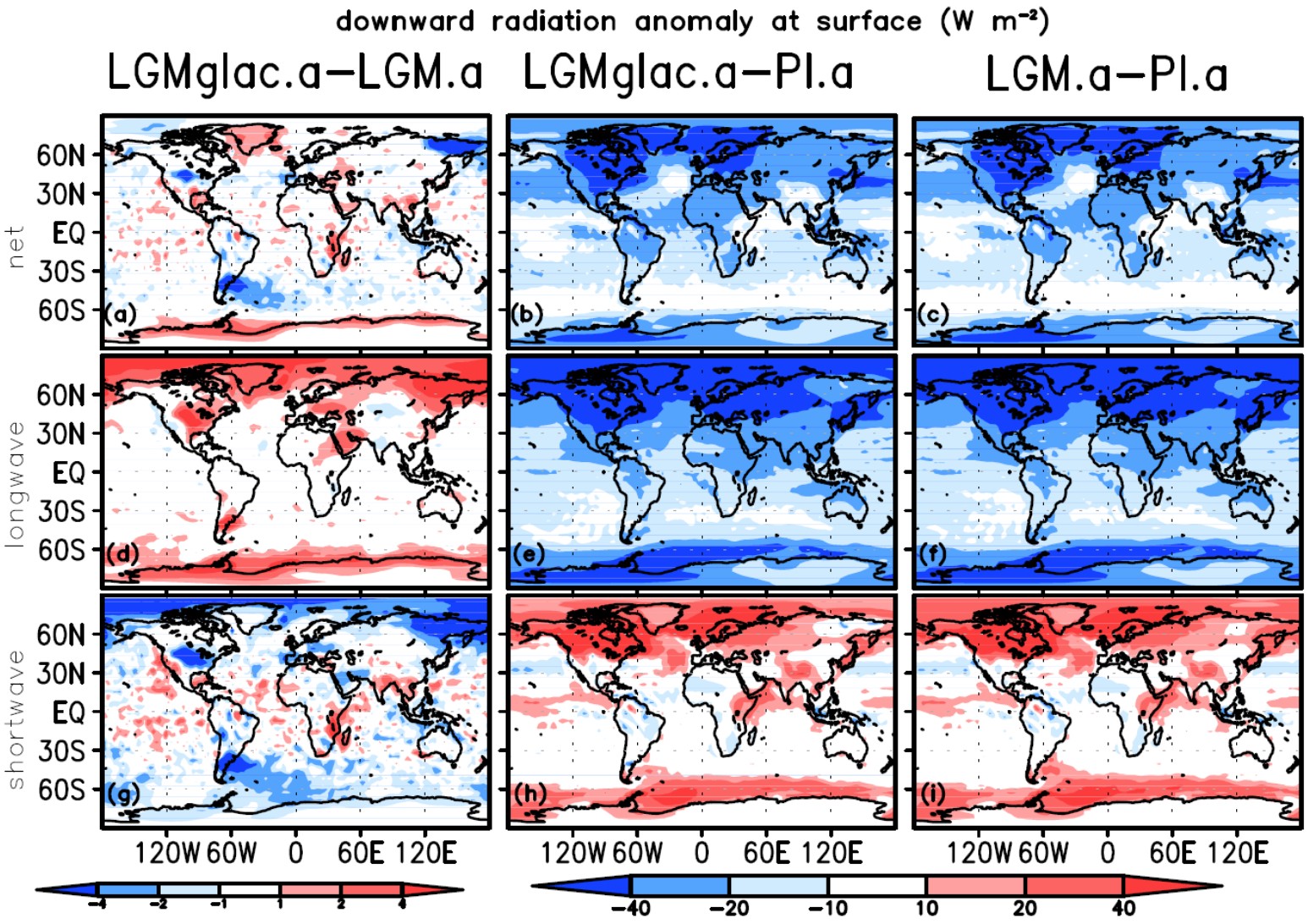

Figure 7: Change in (a) net, (d) longwave, and (g) shortwave downward radiation at the surface LGMglac.a–LGM.a (W m⁻²) (downward, positive). The same (b), (e) and (h) for LGMglac.a-PI.a, and (c), (f) and (i) for LGM.a-PI.a.

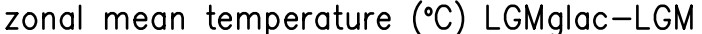

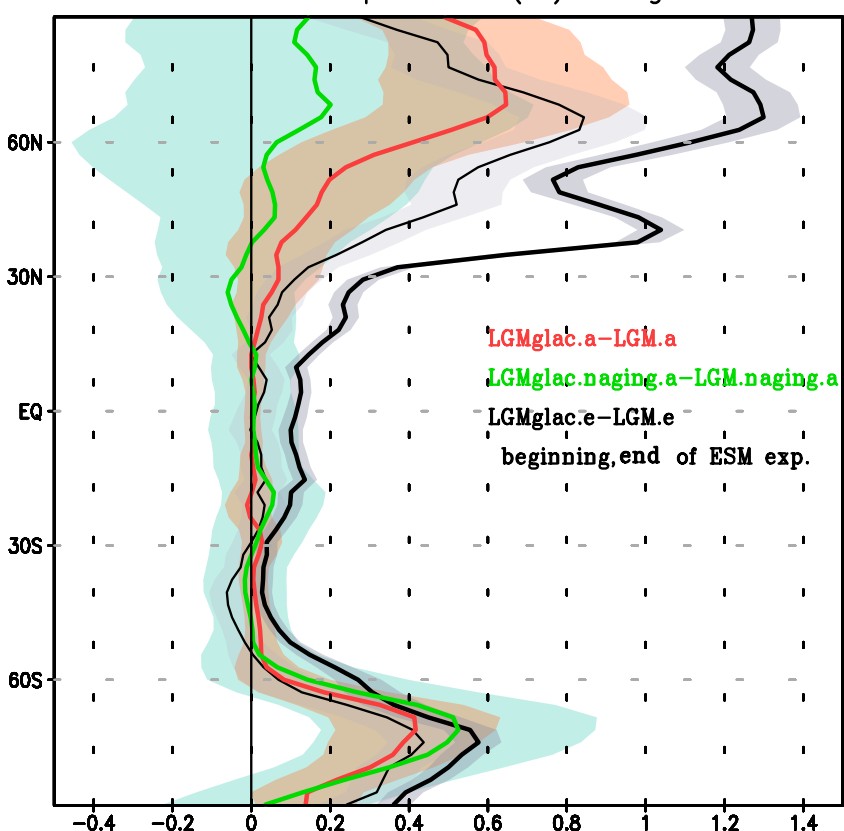

zonal mean temperature (°C) LGMglac−LGM

LGMglac.a−LGM.a

LGMglac.naging.a−LGM.naging.a

LGMglac.e−LGM.e

beginning,end of ESM exp.

Figure 8: Difference in 2 m air temperature between LGMglac and LGM. Red line denotes LGMglac.a–LGM.a. Green line denotes LGMglac.naging.a–LGM.naging.a, which means the change is not attributable to the ageing effect of snow. Thin and thick black lines denote LGMglac.e–LGM.e at the beginning (average of year 1 to 100 in Fig. 1) and the end (average of year 701 to 900) of the experiments, respectively. Shading represents the year-to-year standard deviation.

.

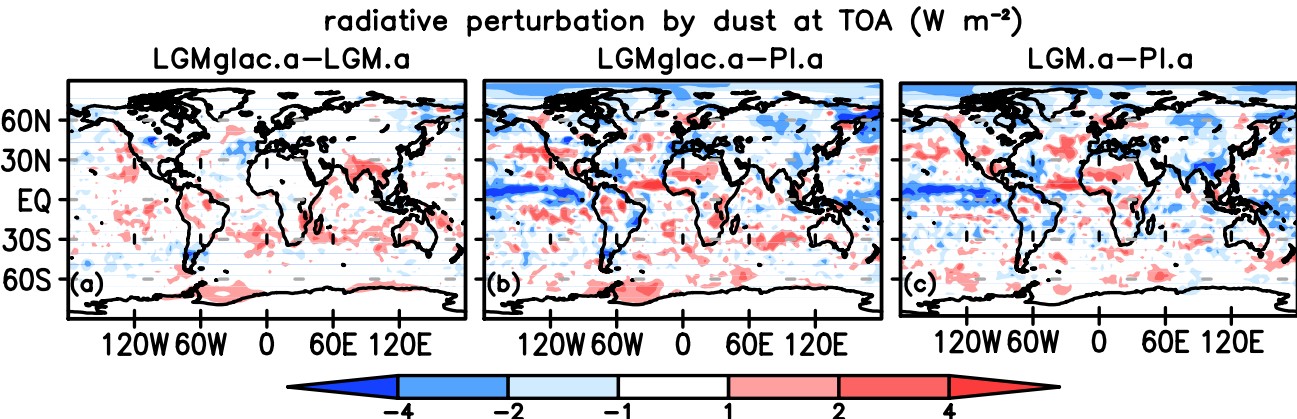

**Fig. 9: Change of net radiative perturbation by dust at the top of the atmosphere (TOA): (a) LGMglac.a–LGM.a, (b) LGMglac.a–PI.a, and (c) LGM.a–PI.a.**

radiative perturbation by dust at surface (W m⁻²)

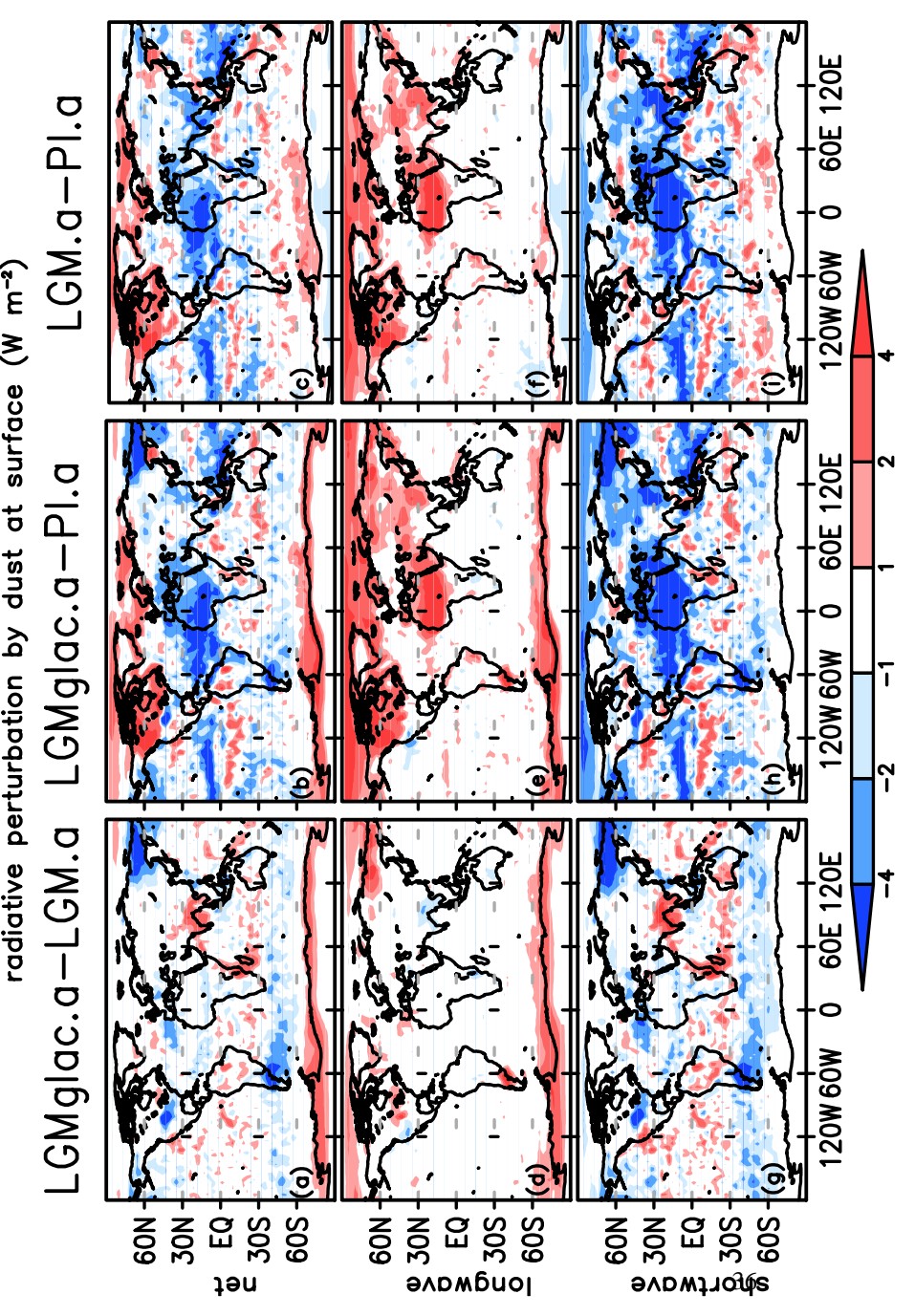

**Figure 10: Change of net radiative perturbation by dust at the surface: (a) LGMglac.a–LGM.a, (b) LGMglac.a–PI.a, and (c) LGM.a–PI.a. Decomposition of net change for the longwave: (d) LGMglac.a–LGM.a, (e) LGMglac.a–PI.a, and (f) LGM.a–PI.a and for the shortwave: (g) LGMglac.a–LGM.a, (h) LGMglac.a–PI.a, and (i) LGM.a–PI.a.**

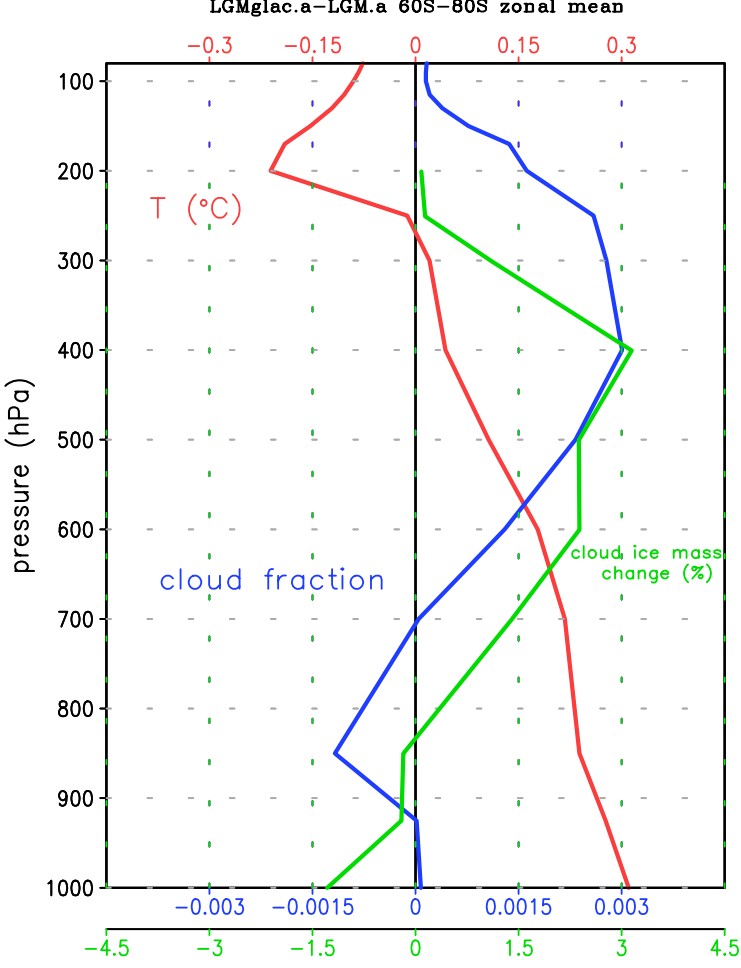

**Figure 11: Averaged value height plot (60°–80° S) for change in LGMglac.a–LGM.a for temperature (red), cloud fraction (blue), and cloud ice mass concentration (green). Note the cloud ice mass concentration is plotted only at values exceeding 1e-8 kg kg⁻¹ in LGM.a.**

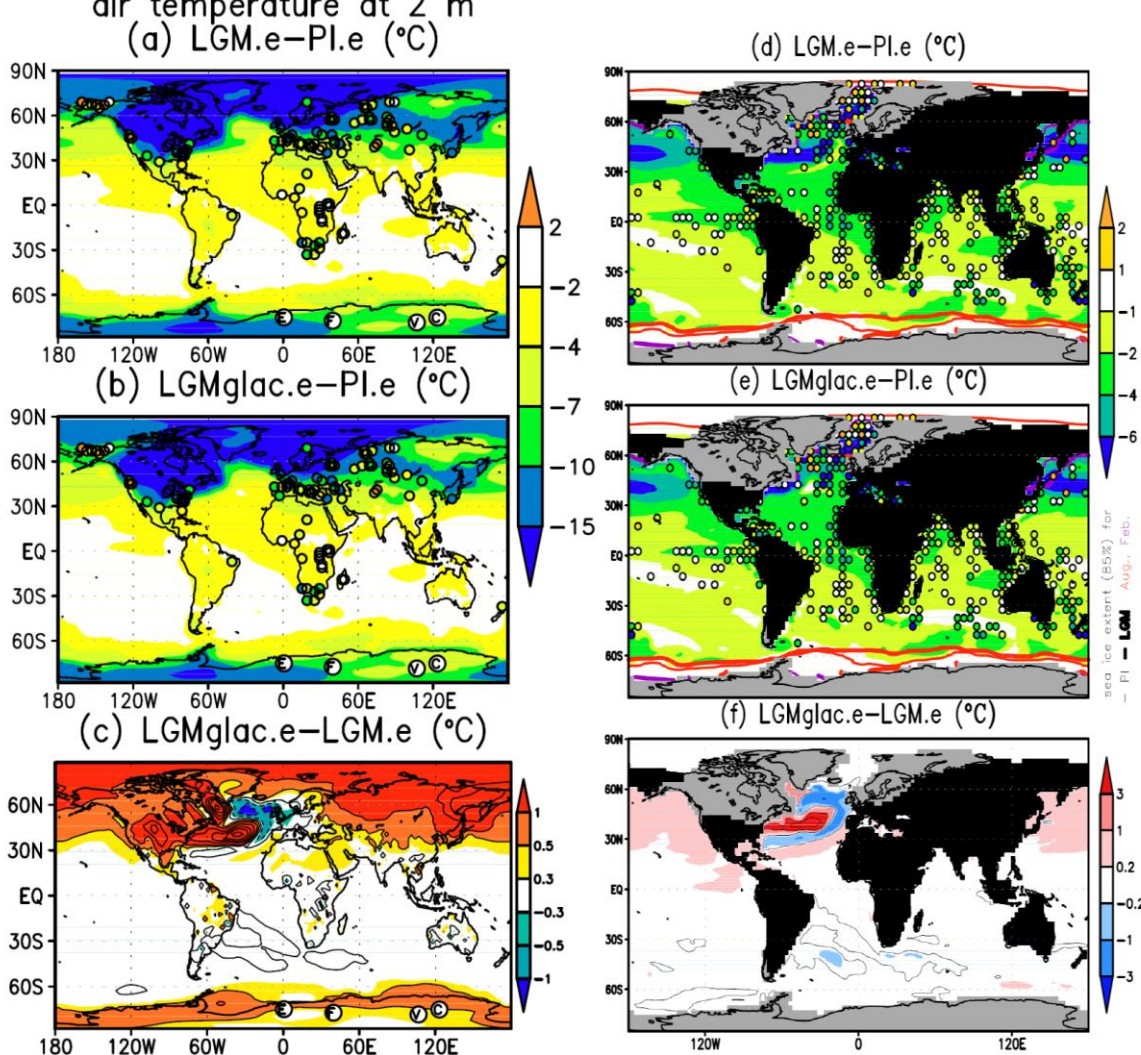

5    **Figure 12: Difference of surface temperature at 2 m height: (a) LGM.e–PI.e, (b) LGMglac.e–PI.e, and (c) LGMglac.e–LGM.e. Coloured circles represent reconstructed temperature change by pollen proxy archives (Bartlein et al., 2011). Circled letters in Antarctica represent four ice core locations: E for EDML, F for Dome Fuji, V for Vostok, and C for Dome C. Sea surface temperature (SST) changes: (d) LGM.e–PI.e, (e) LGMglac.e–PI.e, and (f) LGMglac.e–LGM.e. Purple and red lines in (d) and (e) are 85 % sea ice**

concentration in February and August for PI (thin) and LGM (thick), respectively. Coloured circles represent MARGO SST reconstruction (MARGO project members, 2009). Light grey represents ice sheet areas.

**Table 1: (a) Simulated total dust emissions (Tg y$^{-1}$) and atmospheric burden (Tg) for PI.a, LGM.a, and LGMglac.a**

| Experiment | PI.a | LGM.a | LGMglac.a |
|---|---|---|---|
| Emission | 2540 | 7250 | 13400 |
| Burden | 11.09 | 30.65 | 39.20 |

**(b) Glaciogenic dust flux (Tg y$^{-1}$) (Mahowald et al. 2006a) from the areas shown in Supplementary Fig. A in longitudinal order**

| area | Glaciogenic dust flux (Tg y-1) |
|---|---|
| Europe | 288 |
| Eastern Siberia | 3320 |
| Alaska | 39 |
| Western North America | 17 |
| Central North America | 841 |
| Misissippi river basin | 92 |
| Pampas | 1935 |

**Table 2: List of experiments**
**(a) Experiment using MIROC-ESM**

| Experiment names | Explanation | Integration length (years) |
|---|---|---|
| PI.e | The piControl experiment submitted to CMIP5 | 530 |
| LGM.e | The lgm experiment submitted to CMIP5/PMIP3. The integration is extended further 800 years from the end of PMIP3 period | 1200 |
| LGMglac.e | LGM.e + adding glaciogenic dust flux following Mahowald et al. (2006a) | 940 |

15 **(b) Experiments using AGCM part of MIROC-ESM**

| Experiment names | Explanation | Integration length (years) |
|---|---|---|
| PI.a | Pre-industrial control, SST, sea ice and LAI are taken from the climatology of | 25 |
| LGM.a | The lgm experiment submitted to CMIP5/PMIP3. The integration is extended further 800 years | 25 |

| LGMglac.a | LGM.e + adding glaciogenic dust flux following Mahowald et al. (2006a) | 25 |
|---|---|---|
| LGM.naging.a | LGM.a + no ageing of snow albedo | 25 |
| LGMglac.naging.a | LGMglac.a + no ageing of snow albedo | 25 |

**Table 3: LGMglac.a–PI.a and LGM.a–PI.a changes in global mean radiative perturbation by dust: (a) at the surface and (b) at the top of the atmosphere (TOA) (W m$^{-2}$)**

| (a)  surface | LGMglac.a-PI.a Aerosol-radiation | LGM.a-PI.a Aerosol-radiation | LGMglac.a-PI.a Aerosol-cloud | LGM.a-PI.a Aerosol-cloud |
|---|---|---|---|---|
| net | -0.30 | -0.21 | -0.42 | -0.28 |
| Long wave | 0.37 | 0.28 | 0.50 | 0.34 |
| Short wave | -0.67 | -0.50 | -0.92 | -0.62 |

| (b)  TOA | LGMglac.a-PI.a Aerosol-radiation | LGM.a-PI.a Aerosol-radiation | LGMglac.a-PI.a Aerosol-cloud | LGM.a-PI.a Aerosol-cloud |
|---|---|---|---|---|
| net | 0.12 | 0.07 | -0.39 | -0.36 |
| Long wave | 0.17 | 0.14 | 0.62 | 0.26 |
| Short wave | -0.05 | -0.07 | -1.01 | -0.63 |