# Peer review of "Effect of high dust amount on surface temperature during the Last Glacial Maximum: A modelling study using MIROC-ESM"

_Climate of the Past, 2018_

## Referee Comment (RC1) · Anonymous Referee #1 · 6 Mar 2018

Ohgaito et al., present a comprehensive set of simulations with atmosphere-only and coupled GCM simulations of the last glacial maximum dust cycle. They include glaciogenic dust, dust-cloud interactions and dust-cryosphere interactions. Their study is thorough and potentially the most complete analysis of this problem to date and provides fundamental new insight into the role of aerosols in LGM climate. This would be extremely useful for upcoming PMIP/CMIP5 analyses. The results are interesting and thought provoking, especially as they seem to partly contradict the basic premise that more dust enhances cooling.

**General comments**

[Figure]

Overall, I didn't find this work is placed very well in the context of past studies. How does the dust-cloud scheme used differ from Takemura et al 2009, and Sagoo et al 2017? How comparable is the snow-ageing scheme to Krinner et al 2006, or Ganopolski et al, 2010? Please re-write the introduction to better place the current work in the context of past studies. What is different (or the same) as past work? What do you hope to find? What are main uncertainties etc?

The manuscript has insufficient detail on the methods used, especially on how glaciogenic dust was included. Did you tune the fluxes to the LGM dust observations somehow?

How well does your snow ageing model agree with other schemes (e.g Warren Wiscombe, 1980). Are your LGM results comparable with e.g. Krinner et al 2006?

You do not include any discussion of potential uncertainties, which would seem to be quite large, especially for dust-cloud interactions. Perhaps summarise the approach in SPRINTARS compared to other models (e.g. Komurcu et al., 2014). Are your dust-cloud effects in agreement with those presented for e.g. 'high dust' by Sagoo et al 2017? If not, could you speculate as to why.

Please also could you explain why the dust-cloud effects are so important in the southern hemisphere, but not in the northern hemisphere, and also why the reverse is true for the snow-ageing. Could you expand figure 9 to compare the radiative perturbations from the 3 separate effects of dust that you have studied.

Hence, I would recommend major revisions to the text before publication.

**Specific comments**

Page 3, lines 3 to page 4 line 2. This whole section could be summarised more succinctly for the reader. What is the main message from all previous work? What were the main steps? I would say, most studies simulate a cooling effect, but it is variable and that the introduction of (i) vegetation feedback (Mahowald et al 1999), and (ii) glaciogenic sources (Mahowald et al 2006) and (iii) dust-cloud interactions (Takemura et al 2009, Sagoo et al 2017) are the main developments.

Page 4: Lines 3-11. I find it incomplete here to only list the inclusion of the ocean. You should also mention the dust-cloud interactions and the dust on snow effects and the inclusion of glaciogenic sources in this study.

Page 5: lines 3-4: Did you reduce the imaginary part of the dust refractive index as done by Takemura et al 2009 (their page 3063)?

Page 6: Lines 5-6. More detail of the glaciogenic model setup is required. Did you optimise the fluxes from the emissions using the ice-core data, or marine data or both? What simulations did you use to calculate this? Or did you simply scale emissions in these regions to match the emissions simulated by Mahowald et al 2006?

Page 9: lines 13-16: Isn't it more likely that this small 1 degree shift, is showing that the effect is small over North America? Your argument seems to be that a much higher resolution model would be more sensitive, but I can't see why this should necessarily be the case? Perhaps I have misunderstood.

Figure 8: Can I suggest you separate this plot out into several panels for clarity?

Figure 9: It would be nice to compare the dust-radiation, dust-cloud and dust-cryosphere effects somehow?

Table 2: Takemura et al 2009 quote -0.9 Wm-2 for the net dust-cloud effect at the LGM relative to the PI, but your LGM.a -PI.a difference is only -0.36 Wm-2. Could you comment on the differences with that older study?

**Technical comments**

Abstract Line 23: "for a first trial": I think you are referring to coupling with the ocean? It might make more sense to say "for testing the dust feedbacks in a fully coupled GCM for the first time" or similar?

Abstract Line 25: Perhaps change "interaction" to "coupling"?

Page 2 line 17: "Although mineral dust aerosol is not the most significant cause of warming, its effect is not negligible because it is the most abundant aerosol." This makes it sound like mineral dust might have contributed to recent warming. Suggest to rephrase as "Mineral dust is the most abundant natural aerosol today."

Page 3 Line 13: "where supposed to generate substantial amount of moraine debris during glacial periods" Change "where" to "were". Perhaps include some of the primary references on this topic.

Page 4: Line 4: "The feedback of the aerosol to the ocean and sea ice and back to the atmosphere was not taken into account". Technically, in a slab ocean model the sea-ice can respond, only the oceanic circulation is fixed.

Page 4: Line 19: So the vegetation is not fully dynamic?

Page 5 Line 6: "that control" not "correlated to the"

Also, do these variables also control the glaciogenic dust flux?

Page 6 line 10: "The emission area is also consistent between the experiments, with little deviation following the land-sea mask of MIROC-ESM" Sorry, I don't follow this.

Page 7, line1: Is it really drier over the Sahara? I would be less surprised if it was stronger winds?

Page 7: line 3: "is probably because of the increased soil moisture, resulting in an enhancement of precipitation" Shouldn't this be "resulting from"?

Page 7 line 21: change "location" to "source".

Page 8 line 10: "It represents the total effect of the glaciogenic dust on radiation towards the earth surface" Do you mean dust-radiation plus dust-cloud plus dust-cryosphere interactions?

Page 8 line 19: Repeated sentence.

Page 9 line 7: Refer to figure 6 here.

Page 9: 18-19: Please can you briefly summarise what these are?

Page 10 line 16: i.e. it contributes to atmospheric heating.

Page 13 line 12: "draught" should be "drought".

Page 15 line 15-16: How strong is this snow bias in MIROC-ESM? Might be worth shoing

Figure 8: This caption doesn't completely make sense to me: "Green line denotes LGMglac.naging.a-LGM.naging.a, which means the change arose from non-aging effect of snow albedo." Does this mean that the snow albedo is affected by dust but not by ageing?

Also change "Shades" to "Shading".

––––––––––––––––––––––––––––

---

## Referee Comment (RC2) · Anonymous Referee #2 · 9 Mar 2018

The manuscript by Ohgaito and colleagues presents results of a study on the impacts of dust on the Last Glacial Maximum climate, conducted with different configurations of the MIROC-ESM global model. Attention is given to the role of glaciogenic source of dust. Dust feedbacks on climate include direct effects, cloud effects, and snow darkening. Dust effects are discussed, in terms of perturbation to the atmospheric radiation budgets and surface temperatures. The study is an interesting contribution to both the dust community and the paleoclimate community. In my opinion the manuscript still need some improvement before publication.

General comments

The abstract seems all focused on glaciogenic dust, whereas the title and the manuscript deal with both glaciogenic and non-glaciogenic dust. I suggest to make more clear in the abstract that both aspects are analyzed, and what are the relative contributions to the net dust effects.

A more detailed description of what glaciogenic sources represent, and how glaciogenic sources are embedded in the model setup are strongly encouraged, given the relevant role they play in this manuscript.

The discussion should be improved by comparing more extensively with existing results from the literature, and by enhancing the last section which is an original contribution.

Specific comments

1/14: "the impact of glaciogenic dust". Do you mean "glacial climate dust"? In fact your study explores the effect of both glaciogenic and non-glaciogenic dust.

1/18: "sources" rather than "provenances"

1/21: one gets curious here: is the enhanced cloud cover caused by semi-direct or indirect effects?

1/22-23: It's not clear what you mean by "a first trial of glacial dust modelling" in the specific context of fully-coupled simulations, rather than the atmosphere-only ones.

2/7-8: Rather than "capturing past climate sensitivity", I would say "estimating climate sensitivity by looking at past climates", or perhaps more appropriate for the scope of this manuscript, "capture past climate conditions".

2/21-3/2: Repetition that higher dust fluxes are more pronounced at higher latitudes

Pages 3-4: In this historical review section some recent, relevant papers are not cited, e.g. Albani et al. (2014), Sagoo and Storevlmo (2017). I would recommend to consider them along with other also studies in the discussion section, in terms of global dust budget and impacts.

3/11-13: This sentence is not grammatically correct, please rephrase. Also, moraine debris does not appear to be itself a potential dust source type, but rather fine grained material would be. Please try to be more specific in your definition of glaciogenic sources (e.g. see Bullard et al. 2016).

4/9-10: it's not very clear what is the difference between Sections 3.2. and 3.3 in this brief description.

5/11-13: I do not understand this sentence, i.e. how this weighting occurs

5/14-15: Are you using this kind of off-line model in this study? If not, it seems irrelevant yo mention this fact here.

5/20: Maybe "specific" rather than "particular" would be more appropriate here?

6/9-10: How is this implemented in the model? At the level of grid cells (do you have the same horizontal grid?)? Or rather you are redistributing total emissions on your own grid cells matching the spatial coverage of the same geographical area? Are the emission fluxes prescribed as a repeated monthly varying quantity, or some other way? Please provide more details on this central part of your methodology, and list the geographical location of these glaciogenic sources.

Table 1: Does the integration length refer to the length of your simulations only, or does it also correspond to the period averaged to derive the diagnostic quantities discussed and compared in the manuscript? Please specify how long was our spin-up and how many years you averaged for analysis.

6/18-19: Indeed Australia is the major missing dust source, but also South Africa and the SW North America would fall into this category. Can you comment on how the present day simulations with the same model perform in this respect?

7/3-4: "enhancement" is repeated twice

7/5-6: expressing these quantities in Tg/year would help the reader relating to the

existing literature. Actually it would be very useful to report global budgets of dust emissions, load, and deposition in a table.

7/11: What do you mean exactly by "higher uplift"? Transport to higher levels in the troposphere? Please clarify this aspect, as it may be confused with larger emissions (which should not be case, since glaciogenic sources appear to be prescribed to a fix emissions flux).

Figure 4: please specify if the data reported from Albani et al. (2014) refer to the bulk or to the fine fraction in terms of particle size range. In the caption, please change to "Crosses represent terrestrial sediments, circles marine sediment cores, and diamonds ice core data" - terrestrial sediments are typically loess sections.

7/21: Do you mean "the main source of dust deposited in this region"?

8/6: What do you mean by "glacial dust"? Glaciogenic dust or glacial climate dust? IF you mean the second one, it would be useful to explicitly clarify the distinction, better in earlier sections of the manuscript. If not, you should consistently use "glaciogenic" rather than "glacial" to avoid confusion, I think.

8/11: Do you mean "Figure 7 shows a reduction in the shortwave radiation anomaly ..."? Similarly, in the following lines, I would suggest referring to "-wave radiation anomaly".

8/15-16: What do you mean by "radiative perturbation by the dust"? And how is that different from the analysis just carried out in the previous lines?

8/6-16: A comparison with Mahowald et al. (2006) seems in order here, being the only other study discussing directly the impacts of glaciogenic sources.

8/17-19: This paragraph is repeated twice.

9/6: Please indicate where we can see this effect, i.e. "the cooling effect of the dust loading in the atmosphere" - it is not self-evident.

9/6-9: Again, where can the reader see these features?

9/6-16: It would seem appropriate to compare you results for this process at least with the study by Krinner et al. (2006).

9/20: A net cooling of . . . how much?

Table 2: Could you further split aerosol-radiation interactions between snow darkening and atmospheric impacts? Also, can you indicate the total dust radiative perturbation (from all types of feedback)?

10/1-18: There is ample space here to compare the results in terms of aerosol-radiation interactions with additional existing work, e.g. see Albani et al. (2014) and Hopcroft et al. (2015).

10/16-18: Please rephrase, this sentence is not very clear to me.

10/19-20: The link between this statement an Figure 10 is not clear to me. Please review this passage.

11/2-6: A comparison with Sagoo and Storelvmo (2017) would be appropriate here.

11/7: A more precise title for this section could be "Influence of glaciogenic sources on the ocean SST"?

11/7-21: This section is potentially very interesting. Unfortunately in its present form the discussion is quite superficial in my opinion. I would recommend to expand the section and perhaps enhance Figure 12 with a scatterplot or some other representation that would allow the readers to appreciate the effects on SST and land temperature anomalies.

13/1-3: As discussed in the manuscript, the mismatch is to be attributed to the lack of dust emissions in regions such as Australia in the model used for this study. I fail to see what's the link with the prescribed glaciogenic sources.

[Figure]

14/6: see also Mahowald et al. (2014) or Albani et al. (2014)

References

Albani, S., Mahowald, N. M., Perry, A. T., Scanza, R. A., Zender, C. S., Heavens, N. G., Maggi, V., Kok, J. F., and Otto-Bliesner, B. L.: Improved dust representation in the Community Atmosphere Model, J. Adv. Model. Earth Syst., 6, 541–570, doi:10.1002/2013MS000279, 2014.

Sagoo, N., and Storelvmo, T.: Testing the sensitivity of past climates to the indirect effects of dust: Dust Indirect Effects in Past Climates, Geophys Res Lett., 44(11),5807–5817, 2017.

Bullard, J.E., Baddock, M., Bradwell, T., Crusius, J., Darlington, E., Gaiero, D., Gassó, S., Gisladottir, G., Hodgkins, R., McCulloch, R., McKenna Neuman, C., Mockford, T., Stewart, H., Thorsteinsson, T.: High latitude dust in the Earth System, Reviews of Geophysics, 54, doi:20.1002/2016RG000518, 2016.

Mahowald, N., Yoshioka, M., Collins, W., Conley, A., Fillmore, D., and Coleman, D.: Climate response and radiative forcing from mineral aerosols during the last glacial maximum, pre-industrial and doubled-carbon dioxide climates, Geophys. Res. Lett., 33, D10202, doi:10.1029/2006GL026126, 2006.

Krinner, G., Boucher, O., and Balkanski, Y.: Ice-free glacial northern Asia due to dust deposition on snow, Clim. Dynam., 27, 613–625, 2006.

Hopcroft, P.O., Valdes, P.J., Woodward. S. and Joshi, M.: Last glacial maximum radiative forcing from mineral dust aerosols in an Earth System model, Journal of Geophysical Research, 120, 8186-8205, doi:10.1002/2015JD023742, 2015.

Mahowald, N., Albani, S., Kok, J. F., Engelstaeder, S., Scanza, R., Ward, D. S., and Flanner, M. G.: The size distribution of desert dust aerosols and its impact on the Earth system, Aeolian Res., 15, 53–71, doi:10.1016/j.aeolia.2013.09.002, 2014.

---

## Author Response (AR1)

**Responses to Anonymous Referee #1 on "The effect of high dust amount on the surface temperature during the Last Glacial Maximum: A modelling study using MIROC-ESM" by Cp-2018-2 Ohgaito et al.**

We wish to express our appreciation to the referee for the constructive and insightful comments and suggestions, which have helped us
5  improve our manuscript considerably. In the following, the referee's comments are written in black and our replies are written in blue.

**General comments**

Overall, I didn't find this work is placed very well in the context of past studies. How does the dust-cloud scheme used differ from Takemura et al 2009, and Sagoo et al 2017? How comparable is the snow-ageing scheme to Krinner et al 2006, or Ganopolski et al, 2010? Please re-write the
10  introduction to better place the current work in the context of past studies. What is different (or the same) as past work? What do you hope to find? What are main uncertainties etc?

Our aerosol scheme is identical to that of Takemura et al. (2009). Both Takemura et al. (2009) and Sagoo and Strelvmo (2017) implemented parameterizations of interaction between aerosols and ice crystals based on empirically derived formulations following laboratory
15  experiments and observations (i.e., Lohman and Diehl (2006) and DeMott et al. (2015), respectively). The formulations are different but the schemes of Takemura and Sagoo do similar things; both formulate ice nucleation dependent on temperature and aerosol concentration. It should also be noted that the representations of the cloud water phase of climate models are uncertain and all failed to reproduce the amount and distribution of global observations (Komurcu et al. 2014).

20  Concerning the ageing scheme, Krinner et al. (2006) used an ageing scheme based on Warren and Wiscombe (1980) and Wiscombe and Warren (1980) and the MIROC-ESM used that of Yang et al. (1997) based on Warren and Wiscombe (1982). Ganopolski et al. (2010) used simple scaling of albedo reduction with dust flux relationship. This information has been added in the introduction and model description sections.

25  Our main research objective was to elucidate how glaciogenic dust might influence the global climate, especially surface temperature. This has been added in the introduction.

The manuscript has insufficient detail on the methods used, especially on how glaciogenic dust was included. Did you tune the fluxes to the LGM dust observations somehow?

**In this work, as a first step, we forced additional dust emission constantly following the estimate of Mahowald et al. (2006). The source**

5 **areas of glaciogenic dust in the MIROC-ESM are shown in Supplementary Fig. A. The source strengths for these areas are shown in Table 3 for the non-glaciogenic dust (LGM.a) and the non-glaciogenic and glaciogenic dust (LGMglac.a), following Mahowald et al. (2006a).**

How well does your snow ageing model agree with other schemes (e.g Warren Wiscombe, 1980).

10 **The snow ageing scheme of the MIROC-ESM is that of Warren and Wiscombe (1982). A suitable description has been added in the revised manuscript.**

Are your LGM results comparable with e.g. Krinner et al 2006?

15 **Krinner et al. (2006) suggest that the ageing effect of snow prevents formation of permanent snow over eastern Siberia, consistent with our results. An appropriate statement has been added in the revised text.**

You do not include any discussion of potential uncertainties, which would seem to be quite large, especially for dust-cloud interactions. Perhaps summarise the approach in SPRINTARS compared to other models (e.g. Komurcu et al., 2014).

**Yes, we agree the uncertainty of the aerosol–cloud interaction cannot be overlooked. Komurcu et al. (2014) provided an overview of the uncertainty among the major models and they reported wide ranges of uncertainty in both magnitude and spatial distribution; therefore, our results might differ from other schemes. Acknowledgement of this possibility has been added in the discussion section.**

25 Are your dust cloud effects in agreement with those presented for e.g. 'high dust' by Sagoo et al 2017? If not, could you speculate as to why.

**In terms of the global mean, the negative radiative effect of dust is consistent with Sagoo and Strelvmo (2017) and other studies. In the mid- to low latitudes, our results are also consistent with those previous works with regard to cooling. However, in the high latitudes, our**

results of warming via high dust deposition contrasted with their findings. Because Sagoo and Strelmvo (2017) did not conduct a standard LGM experiment (they changed only CO2 and dust from their control experiment), it is not possible to specify a reason for this. However, their "idealized high dust" means that their emission factor is about 3.4 times that of the control experiment, globally, whereas our glaciogenic dust sources are located in the high latitudes. Therefore, it is likely that the influence of regions of glaciogenic dust emission such as the Pampas of South America on surface temperature around Antarctica is more pronounced in our simulation results. This analysis has been added in the discussion section.

Please also could you explain why the dust-cloud effects are so important in the southern hemisphere, but not in the northern hemisphere, and also why the reverse is true for the snow-ageing. Could you expand figure 9 to compare the radiative perturbations from the 3 separate effects of dust that you have studied. Hence, I would recommend major revisions to the text before publication.

Snow ageing in the MIROC-ESM is tuned to fit the observations in Aoki et al. (2006). According to Aoki et al. (2006), it can be considered (approximately) that albedo starts to reduce with snow impurity of ≥10 ppmw. Dust deposition over the northern high latitudes is of the order of 100 g m$^{-2}$ y$^{-1}$, which corresponds to the order of 1000 ppmw. Conversely, dust deposition near Antarctica is about 0.01 g m$^{-2}$ y$^{-1}$, which corresponds to the order of 0.1 ppmw

Glaciogenic dust travels higher into the troposphere in the Southern Hemisphere and it promotes ice nucleation. Additionally, the dust deposition flux of the standard LGM.a is higher than PI.a in the Northern Hemisphere but lower in the Southern Hemisphere. Therefore, the impact of glaciogenic dust might be more efficient in the Southern Hemisphere. This has been explained in Sect. 3.3.

**Specific comments**

Page 3, lines 3 to page 4 line 2. This whole section could be summarised more succinctly for the reader. What is the main message from all previous work? What were the main steps? I would say, most studies simulate a cooling effect, but it is variable and that the introduction of (i) vegetation feedback (Mahowald et al 1999), and (ii) glacio genic sources (Mahowald et al 2006) and (iii) dust-cloud interactions (Takemura et al 2009, Sagoo et al 2017) are the main developments.

The introduction has been rewritten more succinctly following your suggestions.

Page 4: Lines 3-11. I find it incomplete here to only list the inclusion of the ocean. You should also mention the dust-cloud interactions and the dust on snow effects and the inclusion of glaciogenic sources in this study.

**The sentence has been modified according to your suggestions.**

Page 5: lines 3-4: Did you reduce the imaginary part of the dust refractive index as done by Takemura et al 2009 (their page 3063)?

**Our aerosol module (SPRINTARS) is identical to that of Takemura et al. (2009). The refractive index of dust aerosols was taken from Deepak and Gerber (1983), but its imaginary part was reduced for consistency with recent measurements of weaker shortwave absorption.**

Page 6: Lines 5-6. More detail of the glaciogenic model setup is required. Did you optimise the fluxes from the emissions using the ice-core data, or marine data or both? What simulations did you use to calculate this? Or did you simply scale emissions in these regions to match the emissions simulated by Mahowald et al 2006?

15 **Our method is simple. As a first step, to develop a more sophisticated method for obtaining a best fit to the proxy data archive, we specified the area of glaciogenic dust emission (Supplementary Fig. A) and allowed the emission of a constant dust flux following the estimate of Mahowald et al. (2006). The next step will be to introduce a more realistic method for the emission of glaciogenic dust. We intend to investigate this in subsequent research using an updated version of the MIROC model, which is now under preparation for the submission of experiments to PMIP4. Here, we acknowledge that we adopted a simple method but it was shown successful in obtaining better dust**
20 **deposition distribution in comparison with the proxy data. Improvement of the scheme is certainly required; however, we think even if a difference in amplitude is derived, the main conclusion will still hold.**

Page 9: lines 13-16: Isn't it more likely that this small 1 degree shift, is showing that the effect is small over North America? Your argument seems to be that a much higher resolution model would be more sensitive, but I can't see why this should necessarily be the case? Perhaps I have
25 misunderstood.

**We agree that the sentences were confusing and we have rewritten them.**

Figure 8: Can I suggest you separate this plot out into several panels for clarity?

**For clarity, the shading has been changed to be semi-transparent.**

5  Figure 9: It would be nice to compare the dust-radiation, dust-cloud and dustcryosphere effects somehow?

**We have created Supplementary Fig. C. It shows the LGMglac.a–LGM.a anomaly of aerosol–radiation and aerosol–cloud interactions for the TOA and the surface. Furthermore, it also shows the same format without the snow ageing effect. The panels clarify that the snow ageing effect on the radiative perturbation is minor. The figure also clarifies that the anomaly of aerosol–radiation interaction tends to be**

10  **significant at the level of 0.1 W m⁻², whereas the significance of the aerosol–cloud interaction is difficult to determine. Nevertheless, the positive anomaly around Antarctica at the surface is significant.**

Table 2: Takemura et al 2009 quote -0.9 Wm-2 for the net dust-cloud effect at the LGM relative to the PI, but your LGM.a -PI.a difference is only

15  -0.36 Wm-2. Could you comment on the differences with that older study?

**The model of Takemura et al. (2009) and ours both use the SPRINTARS aerosol module. However, there are differences between the experimental setups for PI and LGM experiments and the model version.**
**The difference of the global mean value is derived mainly from the different boundary conditions for PI. The SST used by Takemura et al.**

20  **(2009) (Ohgaito et al. 2009; Fig. 1) over the warm pool is about 1° warmer than the SST used in this study (Sueyoshi et al. 2013; Fig. 4). It suggests different convective activity, resulting in different amounts of cloud ice and cloud water. This tropical difference influences the global mean value, suggesting that the SST bias of the control experiment could affect both regional and global mean values. This discussion has been added in Sect. 4.**

25  **Technical comments**
Abstract Line 23: "for a first trial": I think you are referring to coupling with the ocean? It might make more sense to say "for testing the dust feedbacks in a fully coupled GCM for the first time" or similar?

**Thank you for this observation. It has been changed accordingly.**

Abstract Line 25: Perhaps change "interaction" to "coupling"?

5  **This has been changed as suggested.**

Page 2 line 17: "Although mineral dust aerosol is not the most significant cause of warming, its effect is not negligible because it is the most abundant aerosol." This makes it sound like mineral dust might have contributed to recent warming. Suggest to rephrase as "Mineral dust is the most abundant natural aerosol today."

**This has been changed.**

Page 3 Line 13: "where supposed to generate substantial amount of moraine debris during glacial periods" Change "where" to "were". Perhaps include some of the primary references on this topic.

**The sentence has been changed.**

Page 4: Line 4: "The feedback of the aerosol to the ocean and sea ice and back to the atmosphere was not taken into account". Technically, in a slab ocean model the sea-ice can respond, only the oceanic circulation is fixed.

**The sentence has been rewritten.**

Page 4: Line 19: So the vegetation is not fully dynamic?

25  **The dynamic vegetation module simulates global vegetation dynamics and terrestrial carbon cycling (Sato et al., 2007) using the output of the physical module, but it returns only the LAI and amount of carbon back to the land and atmosphere, respectively. Thus, the dynamic vegetation model is loosely coupled with the MIROC-ESM.**

Page 5 Line 6: "that control" not "correlated to the" Also, do these variables also control the glaciogenic dust flux?

**This has been changed and explanation added regarding glaciogenic dust.**

5   Page 6 line 10: "The emission area is also consistent between the experiments, with little deviation following the land-sea mask of MIROC-ESM" Sorry, I don't follow this.

**Supplementary Fig. A has been added to clarify the source areas of glaciogenic dust used in our experiments and the sentence has been reworded.**

Page 7, line1: Is it really drier over the Sahara? I would be less surprised if it was stronger winds?

**Yes, you are correct. Stronger wind is the reason for more dust from desert areas. The sentence has been modified appropriately in the revised text.**

Page 7: line 3: "is probably because of the increased soil moisture, resulting in an enhancement of precipitation" Shouldn't this be "resulting from"?

**This has been changed accordingly.**

20   Page 7 line 21: change "location" to "source".

**This has been changed accordingly.**

Page 8 line 10: "It represents the total effect of the glaciogenic dust on radiation towards the earth surface" Do you mean dust-radiation plus dust-
25   cloud plus dust-cryosphere interactions?

**We mean the total effect of the glaciogenic dust load in the atmosphere toward the surface of the earth. The sentence has been rewritten to clarify this point.**

Page 8 line 19: Repeated sentence.

**Thank you. The duplicated text has been deleted.**

Page 9 line 7: Refer to figure 6 here.

**We have done as you suggested.**

10   Page 9: 18-19: Please can you briefly summarise what these are?

**An appropriate explanation has been added.**

Page 10 line 16: i.e. it contributes to atmospheric heating.

**The global mean radiative perturbation by glaciogenic dust is cooling ($-0.19$ W m$^{-2}$)**
**However, glaciogenic dust behaves differently over the polar regions and it contributes to atmospheric heating. An appropriate explanation has been added in the revised manuscript.**

20   Page 13 line 12: "draught" should be "drought".

**Thank you for identifying this error; it has been changed accordingly.**

Page 15 line 15-16: How strong is this snow bias in MIROC-ESM? Might be worth shoing

**Supplementary Fig. H has been added to show that snow cover tends to remain in boreal spring over southern Siberia.**

Figure 8: This caption doesn't completely make sense to me: "Green line denotes LGMglac.naging.a-LGM.naging.a, which means the change arose from non-aging effect of snow albedo." Does this mean that the snow albedo is affected by dust but not by ageing? Also change "Shades" to "Shading".

5    **We wanted to say that the "LGMglac.naging.a–LGM.naging.a" shows "the change is not attributable to the ageing effect of snow". The caption for the figure has been rewritten in the revised text.**

**Response for the Anonymous Referee #2 on "The effect of high dust amount on the surface temperature during the Last Glacial Maximum: A modelling study using MIROC-ESM" by Rumi Ohgaito et al.**

The manuscript by Ohgaito and colleagues presents results of a study on the impacts of dust on the Last Glacial Maximum climate, conducted with different configurations of the MIROC-ESM global model. Attention is given to the role of glaciogenic source of dust. Dust feedbacks on climate include direct effects, cloud effects, and snow darkening. Dust effects are discussed, in terms of perturbation to the atmospheric radiation budgets and surface temperatures. The study is an interesting contribution to both the dust community and the paleoclimate community. In my opinion the manuscript still need some improvement before publication.

**We wish to express our appreciation to the referee for the positive and constructive comments and suggestions, which have helped us improve our manuscript considerably. In the following, the referee's comments are written in black and our replies are written in blue.**

General comments
The abstract seems all focused on glaciogenic dust, whereas the title and the manuscript deal with both glaciogenic and non-glaciogenic dust. I suggest to make more clear in the abstract that both aspects are analyzed, and what are the relative contributions to the net dust effects.

**The abstract has been modified to include mention of both non-glaciogenic and glaciogenic dust.**

A more detailed description of what glaciogenic sources represent, and how glaciogenic sources are embedded in the model setup are strongly encouraged, given the relevant role they play in this manuscript.

**An appropriate description has been added in Sect. 2.2 and the source areas of glaciogenic dust are shown in Supplementary Fig. A.**

The discussion should be improved by comparing more extensively with existing results from the literature, and by enhancing the last section which is an original contribution.

**We have improved the discussion section following your suggestion.**

**The final section of the manuscript is not simply an analysis of the original contribution but it also provides an evaluation of the effect of glaciogenic dust on surface temperature. We intended to leave detailed analysis of the oceanic response for subsequent study using ongoing PMIP4 model experiments. However, your suggestion made us realize the interest concerning the oceanic element. Therefore, the oceanic response to different dust fluxes under the conditions of the LGM is more discussed in Sect 3.4.**

Specific comments

1/14: "the impact of glaciogenic dust". Do you mean "glacial climate dust"? In fact your study explores the effect of both glaciogenic and non-glaciogenic dust.

10 **Our focus was on glaciogenic dust. Thus, the differences between scenarios with and without glaciogenic dust were analysed as a priority. However, analyses were also performed regarding scenarios with glaciogenic dust and non-glaciogenic dust. The sentence has been modified accordingly in the revised text.**

1/18: "sources" rather than "provenances"

**This has been changed as suggested.**

1/21: one gets curious here: is the enhanced cloud cover caused by semi-direct or indirect effects?

20 **According to the definition of the IPCC AR5 Chapter 7, the aerosol–cloud interaction does not include semi-direct effects. If semi-direct effects dominate, enhancement of cloud prevents shortwave radiation reaching the earth's surface, whereas the change in longwave radiation causes surface warming in this case.**

1/22-23: It's not clear what you mean by "a first trial of glacial dust modelling" in the specific context of fully-coupled simulations, rather than the
25 atmosphere-only ones.

**This has been changed to "an initial examination of the effect of glaciogenic dust on an oceanic general circulation model"**

2/7-8: Rather than "capturing past climate sensitivity", I would say "estimating climate sensitivity by looking at past climates", or perhaps more appropriate for the scope of this manuscript, "capture past climate conditions".

**This has been changed appropriately.**

2/21-3/2: Repetition that higher dust fluxes are more pronounced at higher latitudes

**Thank you. The repetition has been avoided in the revised text.**

10  Pages 3-4: In this historical review section some recent, relevant papers are not cited, e.g. Albani et al. (2014), Sagoo and Storevlmo (2017). I would recommend to consider them along with other also studies in the discussion section, in terms of global dust budget and impacts.

**Both in the historical review and the discussion sections, the works by Albani et al. (2014) and Sagoo and Storevlmo (2017) are now included. The global dust budget of previous studies is summarized in Table 1 of Hopcroft et al. (2015). They highlighted that the dust amount is**
15  **highly dependent on the model, not only for LGM experiments but also for PI experiments. Our emissions and loadings are listed in Table 3. Our values fall in the middle of the ranges determined by previous studies. However, they are close to those of Takemura et al. (2009) for PI and LGM, probably because the models adopted are from the same model family and use the same aerosol module. The emission of LGMglac is close to that of Mahowald et al. (2006a), most likely because we adopted their glaciogenic dust.**

20  3/11-13: This sentence is not grammatically correct, please rephrase. Also, moraine debris does not appear to be itself a potential dust source type, but rather fine grained material would be. Please try to be more specific in your definition of glaciogenic sources (e.g. see Bullard et al. 2016).

**The sentence has been rewritten and the term "moraine debris" has been changed to "glacial flour" (Bullard et al. 2016).**

25  4/9-10: it's not very clear what is the difference between Sections 3.2. and 3.3 in this brief description.

**Section 3.2 describes the effect of glaciogenic dust on surface temperature. The question of how glaciogenic dust might modulate the surface temperature, especially surrounding Antarctica, is discussed in Sect. 3.3. The text has been rewritten accordingly in the revised manuscript.**

5/11-13: I do not understand this sentence, i.e. how this weighting occurs

**The ageing of snow is implemented following Yang et al. (1997) and tuned to fit the observations by Aoki et al. (2003, 2006). The weighting**
5 **parameters are defined according to the absorbing property of the material. However, this part has now been removed because soot is no**
**longer discussed in this paper.**

5/14-15: Are you using this kind of off-line model in this study? If not, it seems irrelevant yo mention this fact here.

10 **Because we discuss this in Sect. 4, the sentence you have identified has now been deleted.**

5/20: Maybe "specific" rather than "particular" would be more appropriate here?

**This has been changed as suggested.**

6/9-10: How is this implemented in the model? At the level of grid cells (do you have the same horizontal grid?)? Or rather you are redistributing
total emissions on your own grid cells matching the spatial coverage of the same geographical area? Are the emission fluxes prescribed as a repeated
monthly varying quantity, or some other way? Please provide more details on this central part of your methodology, and list the geographical
location of these glaciogenic sources.

**The glaciogenic source areas are defined by following Mahowald et al. (2006). Supplementary Fig. A has been added to clarify the source**
**areas of glaciogenic dust. For each source area, we set a constant dust emission to match the flux in Mahowald et al. (2006). As a first trial,**
**glaciogenic dust is emitted constantly. Once it emitted, the treatment of the dust is the same as any other dust, i.e., its transportation,**
**advection, and deposition processes. Although constant emission cannot happen in nature, this attempt was simply intended to emit the**
25 **identical flux as in Mahowald et al. (2006) as a first step. Introducing temporal variation in emission and obtaining original glaciogenic**
**dust flux that fits the updated proxy archive is the next research ambition. This has been outlined in Sect. 4.**

Table 1: Does the integration length refer to the length of your simulations only, or does it also correspond to the period averaged to derive the diagnostic quantities discussed and compared in the manuscript? Please specify how long was our spin-up and how many years you averaged for analysis.

5   **The listed integration lengths include the analyses periods. Now the ranges of the analyses are shown in Fig. 1.**

6/18-19: Indeed Australia is the major missing dust source, but also South Africa and the SW North America would fall into this category. Can you comment on how the present day simulations with the same model perform in this respect?

10   **Our PI.a (PI.e) has wet bias and relevant high LAI over South Africa and SW North America. The manuscript has been rewritten to include mention of these areas.**

7/3-4: "enhancement" is repeated twice

15   **Thank you for noticing this error; it has been corrected.**

7/5-6: expressing these quantities in Tg/year would help the reader relating to the existing literature. Actually it would be very useful to report global budgets of dust emissions, load, and deposition in a table.

20   **The unit has been changed and the additional information requested is now presented in Table 3.**

7/11: What do you mean exactly by "higher uplift"? Transport to higher levels in the troposphere? Please clarify this aspect, as it may be confused with larger emissions (which should not be case, since glaciogenic sources appear to be prescribed to a fix emissions flux).

25   **We apologize for the confusion. We meant to indicate greater dust concentration at higher levels of the troposphere. The wording has been changed appropriately in the revised manuscript.**

Figure 4: please specify if the data reported from Albani et al. (2014) refer to the bulk or to the fine fraction in terms of particle size range. In the caption, please change to "Crosses represent terrestrial sediments, circles marine sediment cores, and diamonds ice core data" - terrestrial sediments are typically loess sections.

5    **We used the bulk values of Albani et al. (2014); the caption has been changed accordingly.**

7/21: Do you mean "the main source of dust deposited in this region"?

**Yes, you are correct. We have clarified this in the revised text.**

8/6: What do you mean by "glacial dust"? Glaciogenic dust or glacial climate dust? IF you mean the second one, it would be useful to explicitly clarify the distinction, better in earlier sections of the manuscript. If not, you should consistently use "glaciogenic" rather than "glacial" to avoid confusion, I think.

15    **This was an error. The word has now been changed to "glaciogenic" and the entire manuscript has been checked to avoid other such occurrences.**

8/11: Do you mean "Figure 7 shows a reduction in the shortwave radiation anomaly . . ."? Similarly, in the following lines, I would suggest referring to "-wave radiation anomaly".

**This has been changed appropriately.**

8/15-16: What do you mean by "radiative perturbation by the dust"? And how is that different from the analysis just carried out in the previous lines?

**In this section, we discuss the surface radiation anomaly. In the following section, we discuss the causes of this anomaly. It is clear that the anomaly is caused by glaciogenic dust based on the experimental setting; however, we have separated the effects of aerosol–radiation, aerosol–cloud interactions. A suitable explanation has been added in the text.**

8/6-16: A comparison with Mahowald et al. (2006) seems in order here, being the only other study discussing directly the impacts of glaciogenic sources.

5 **Comparison with Mahowald et al. (2006b) and further discussions have been added.**

8/17-19: This paragraph is repeated twice.

**Thank you for identifying this error. The duplicate text has been deleted.**

9/6: Please indicate where we can see this effect, i.e. "the cooling effect of the dust loading in the atmosphere" - it is not self-evident.

**The likely cooling effect of dust on the earth's surface is suggested in the IPCC AR5 Sec. 7 and references therein. However, the uncertainty ranges from negative to positive. Each of our experiments also resulted in a cooling effect of dust in the global mean (PI.a: −0.99 W m⁻²,**
15 **LGM.a: −1.50 W m⁻², and LGMglac.a −1.71 W m⁻²) at the surface.**

9/6-9: Again, where can the reader see these features?

**Supplementary Fig. B has been added to show the albedo difference between LGMglac.a and LGM.a. A description of Supplementary Fig.**
20 **B has also been added in the revised text.**

9/6-16: It would seem appropriate to compare you results for this process at least with the study by Krinner et al. (2006).

**Thank you for your suggestion. The result of Krinner et al. (2006) is consistent with ours and a sentence explaining this has been added in**
25 **the manuscript.**

9/20: A net cooling of . . . how much?

**Quantification of the cooling has been added in the revised text, i.e., PI.a: −0.99 W m$^{-2}$, LGM.a: −1.50 W m$^{-2}$, and LGMglac.a: −1.71 W m$^{-2}$.**

Table 2: Could you further split aerosol-radiation interactions between snow darkening and atmospheric impacts? Also, can you indicate the total dust radiative perturbation (from all types of feedback)?

**We have created Supplementary Fig. C. It shows the LGMglac.a–LGM.a anomaly of aerosol–radiation and aerosol–cloud interactions for the TOA and the surface. Furthermore, it also shows the same format without the snow ageing effect. The panels clarify that the snow ageing effect on the radiative perturbation is minor. The figure also clarifies that the anomaly of aerosol–radiation interaction tends to be significant at the level of 0.1 W m$^{-2}$, whereas the significance of the aerosol–cloud interaction is difficult to determine. Nevertheless, the positive anomaly around Antarctica at the surface is significant.**

10/1-18: There is ample space here to compare the results in terms of aerosol-radiation interactions with additional existing work, e.g. see Albani et al. (2014) and Hopcroft et al. (2015).

**Comparison with the works of Albani and Hopcroft has now been included and appropriate discussion has been added.**

10/16-18: Please rephrase, this sentence is not very clear to me.

**The sentence has been rephrased appropriately.**

10/19-20: The link between this statement an Figure 10 is not clear to me. Please review this passage.

**The sentence explains the content of Fig. 10. It has been rephrased accordingly.**

11/2-6: A comparison with Sagoo and Storelvmo (2017) would be appropriate here.

**Comparison with Sagoo and Salmiento (2017) and appropriate discussions have been added at the end of paragraph.**

11/7: A more precise title for this section could be "Influence of glaciogenic sources on the ocean SST"?

**Because consideration of the effect of dust on oceans has been added, the section title has been left unchanged.**

11/7-21: This section is potentially very interesting. Unfortunately in its present form the discussion is quite superficial in my opinion. I would recommend to expand the section and perhaps enhance

**We intended to elucidate the oceanic response in our next study using LGM experiments for PMIP4. However, additional analyses have been performed and the findings are explained in the revised text.**

Figure 12 with a scatterplot or some other representation that would allow the readers to appreciate the effects on SST and land temperature anomalies.

**The temperature anomaly of the zonal mean over land and scatter plots of the anomaly of the proxy data and of the anomaly of the corresponding model grids are shown in Supplementary Fig. E. It illustrates the level of agreement between the model and the proxy archives. Pronounced discrepancy is evident in the northern high latitudes with some proxy data suggesting warmer temperatures than PI, whereas the model shows a negative anomaly. Compared with LGM.e, LGMglac.e generally exhibits slightly closer agreement with the proxy data.**

13/1-3: As discussed in the manuscript, the mismatch is to be attributed to the lack of dust emissions in regions such as Australia in the model used for this study. I fail to see what's the link with the prescribed glaciogenic sources.

**We meant that Mahowald et al. (2006a) used the DIRTMAP dust deposition archive (Kohfeld and Harrison 2001) to fit the model deposition flux, which had no proxy points over the southern Pacific Ocean. This could also be one of the reasons for the underestimation. The manuscript has been rewritten to clarify this point.**

14/6: see also Mahowald et al. (2014) or Albani et al. (2014)

**Thank you for your suggestion. These studies have now been cited because discussion of their findings is appropriate in this section of our manuscript.**

*Manuscript showing all the changes from CPD

[revised manuscript text omitted]

LGM.a/PI.a, (b)
for LGMglac.a-
is...how the dus
is ...he tempera

deposition flux (g/m²/y)

[Figure]

**Figure 4:** Model–data comparison of dust deposition flux (g m$^{-2}$ y$^{-1}$) estimated from the ice and sediment core data archives obtained from Kohfeld et al. (2013) and bulk data from Albani et al. (2014): (a) PI.a, (b) LGM.a, and (c) LGMglac.a. Model–data scatter plots for (d) PI.a, (e) LGM.a, and (f) LGMglac.a. Colours and marks represent areas and core types, i.e., red: Eurasia, brown: North America, orange: Indian Ocean, pink and light blue: Atlantic and Pacific oceans in the Northern Hemisphere, respectively, green and light green: Atlantic and Pacific oceans in the Southern Hemisphere, respectively, blue: Southern Ocean, turquoise blue: Arctic, and dark blue: Antarctica. Crosses, circles, and diamonds represent terrestrial, marine core, and ice core sediments, respectively.

[Figure]

**Figure 5:** Model–data comparison of ratio of dust deposition flux estimated from the ice and sediment core data archives obtained from Kohfeld et al. (2013) and Albani et al. (2014): (a) LGM.a/PI.a and (b) LGMglac.a/PI.a.

[Figure]

**Figure 6:** Difference of surface temperature at 2 m height for LGMglac.a–LGM.a. Change is considered not significant at the 95 % confidence level in the hatched area based on a t-test.

[Figure]

**Figure 7:** Change in (a) net, (b) longwave, and (c) shortwave downward radiation at the surface LGMglac.a–LGM.a (W m$^{-2}$) (downward, positive).

[Figure]

**Figure 8:** **Difference in** 2 m air temperature between LGMglac and LGM. Red line denotes LGMglac.a–LGM.a. Green line denotes LGMglac.naging.a–LGM.naging.a, which means the change **is not attributable to the ageing** effect of snow. Thin and thick black lines denote LGMglac.e–LGM.e at the beginning (average **of year 1 to 100** in **Fig.** 1) and the end (average **of year 701 to 900**) of the experiments, respectively. **Shading represents** the year-to-year standard deviation.

.

[Figure]

**Fig. 9: Change of net radiative perturbation by dust at the top of the atmosphere (TOA): (a) LGMglac.a–LGM.a, (b) LGMglac.a–PI.a, and (c) LGM.a–PI.a.**

[Figure]

radiative perturbation by dust at surface (W m⁻²)

**Figure 10: Change of net radiative perturbation by dust at the surface: (a) LGMglac.a–LGM.a, (b) LGMglac.a–PI.a, and (c) LGM.a–PI.a. Decomposition of net change for the longwave: (d) LGMglac.a–LGM.a, (e) LGMglac.a–PI.a, and (f) LGM.a–PI.a and for the shortwave: (g) LGMglac.a–LGM.a, (h) LGMglac.a–PI.a, and (i) LGM.a–PI.a.**

for… (a) LGMg
LGM.a-…PI.a.
longwave for…
and (f) LGM.a-.
LGMglac.a-…L

[Figure]

**Figure 11:** Averaged value height plot (60°–80° S) for change in LGMglac.a–LGM.a for temperature (red), cloud fraction (blue), and cloud ice mass concentration (green). Note the cloud ice mass concentration is plotted only at values exceeding 1e-8 kg kg$^{-1}$ in LGM.a.

[Figure]

change in LGM
(red,…, cloud fr
concentration in
concentration is
exceeding 1e-8 k

5 **Figure 12:** Difference of surface temperature at 2 m height: (a) LGM.e–PI.e, (b) LGMglac.e–PI.e, and (c) LGMglac.e–LGM.e. Coloured circles represent reconstructed temperature change by pollen proxy archives (Bartlein et al., 2011). Circled letters in Antarctica represent four ice core locations: E for EDML, F for Dome Fuji, V for Vostok, and C for Dome C. Sea surface temperature (SST) changes: (d) LGM.e–PI.e, (e) LGMglac.e–PI.e, and (f) LGMglac.e–LGM.e. Purple and red lines in (d) and (e) are 85 % sea ice concentration in February and August for PI (thin) and LGM (thick), respectively. Coloured circles represent MARGO SST

for… (a) LGM.e
LGMglac.e-…L
reconstructed te
(Bartlein et al., 2
Antarctica repr
EDML, F for Do
Sea surface tem
(e) LGMglac.e-…
The…Purple an
concentration in
(thick), respecti

**reconstruction (MARGO project members, 2009). Light grey represents ice sheet areas.**

**Table 1: List of experiments**

**(a) Experiment using MIROC-ESM**

| Experiment names | Explanation | Integration length (years) |
|---|---|---|
| PI.e | The piControl experiment submitted to CMIP5 | 530 |
| LGM.e | The lgm experiment submitted to CMIP5/PMIP3. The integration is extended further 800 years from the end of PMIP3 period | 1200 |
| LGMglac.e | LGM.e + adding glaciogenic dust flux following Mahowald et al. (2006a) | 940 |

**(b) Experiments using AGCM part of MIROC-ESM**

| Experiment names | Explanation |
|---|---|
| PI.a | Pre-industrial control, SST, sea ice and LAI are taken from the climatology of |
| LGM.a | The lgm experiment submitted to CMIP5/PMIP3. The integration is extended further 800 years |
| LGMglac.a | LGM.e + adding glaciogenic dust flux following Mahowald et al. (2006a) |
| LGM.naging.a | LGM.a + no ageing of snow albedo |
| LGMglac.naging.a | LGMglac.a + no ageing of snow albedo |

**Table 2: LGMglac.a-PI.a and LGM.a-PI.a changes in global mean radiative perturbation by dust: (a) at the surface and (b) at the top of the atmosphere (TOA) (W m$^{-2}$)**

| (a) surface | LGMglac.a-PI.a Aerosol-radiation | LGM.a-PI.a Aerosol-radiation | LGMglac.a-PI.a Aerosol-cloud | LGM.a-PI.a Aerosol-cloud |
|---|---|---|---|---|
| net | -0.30 | -0.21 | -0.42 | -0.28 |
| Long wave | 0.37 | 0.28 | 0.50 | 0.34 |
| Short wave | -0.67 | -0.50 | -0.92 | -0.62 |

| (b) TOA | LGMglac.a-PI.a Aerosol-radiation | LGM.a-PI.a Aerosol-radiation | LGMglac.a-PI.a Aerosol-cloud | LGM.a-PI.a Aerosol-cloud |
|---|---|---|---|---|
| net | 0.12 | 0.07 | -0.39 | -0.36 |
| Long wave | 0.17 | 0.14 | 0.62 | 0.26 |
| Short wave | -0.05 | -0.07 | -1.01 | -0.63 |

**Table 3: (a) Simulated total dust emissions (Tg y$^{-1}$) and atmospheric burden (Tg) for PI.a, LGM.a, and LGMglac.a**

| Experiment | PI.a | LGM.a | LGMglac.a |
|---|---|---|---|
| Emission | 2540 | 7250 | 13400 |
| Burden | 11.09 | 30.65 | 39.20 |

**(b) Glaciogenic dust flux (Tg y$^{-1}$) (Mahowald et al. 2006a) from the areas shown in Supplementary Fig. A in longitudinal order**

| area | Glaciogenic dust flux (Tg y-1) |
|---|---|
| Europe | 288 |
| Eastern Siberia | 3320 |
| Alaska | 39 |
| Western North America | 17 |
| Central North America | 841 |
| Misissippi river basin | 92 |
| Pampas | 1935 |

---

## Author Response (AR2)

**Response for the Anonymous Referee #1 on "The effect of high dust amount on the surface temperature during the Last Glacial Maximum: A modelling study using MIROC-ESM" by Rumi Ohgaito et al.**

We wish to express our appreciation for the referee for the constructive comments and suggestions. We modified the manuscript following your suggestions one by one.

The referee's comments are written in black and our replies are written in characters in blue color.

I realise figures 6 & 7 were in the first submission, but I think they need the LGM.a – PI.a anomalies. This would make it consistent with figures 2-5 and 9-12 and the rest of the paper.

LGMglac.a-PI.a and LGM.a-PI.a maps are added in Figure 6 and 7 because we agree that anomaly from PI is one of important interests. Descriptions are added in the manuscript.

It would also help to redress the emphasis from analysing the glaciogenic effect alone, to a more general paper about LGM dust, which is what the introduction leads the reader to expect, and which was commented on by the other reviewer in the first round. This would also make sense given the artificial nature of the glaciogenic dust inputs used.

We agree your suggestion. We renamed Sec. 3.2 and discussion about general LGM-PI anomaly is added.

I now realise I don't understand the difference between Figure 10 (Change of net radiative perturbation by dust at the surface) and figure 7 (Change in (a) net, (b) longwave, and (c) shortwave downward radiation at the surface).

Figure 10 represents that the radiative perturbation induced by dust (aerosol-radiation and aerosol-cloud interactions), whereas Figure 7 shows that the anomaly of downward radiation at the surface, not only by dust but also by all changes, cooled air temperature, humidity, cloud, precipitation, circulation and so on. But the similarity of Figure 7 (a, d, g) and Figure 10 (a, d, g) suggests clearly that the radiation anomaly between LGMglac.a and LGM.a stems from aerosol-radiation + aerosol-cloud interactions of glaciogenic dust.

Page 13, lines 13-19: Is there a longwave aerosol-cloud effect in the northern hemisphere also? E.g. looking at figure 7b or 10e? Otherwise, I don't understand why this only occurs in the southern hemisphere. From figure 7c and 7a, is it possible that in the northern hemisphere it is offset by a short-wave effect, hence it doesn't show up in figure C?

5  There is long wave perturbation as seen in Figure 10 (e). But in the northern high latitudes, the amplitude is comparable to Figure 10 (f). One of the reasons would be the dust load enhanced already in LGM.a in the northern hemisphere relative to PI.a whereas reduced in the southern hemisphere. By adding glaciogenic dust, dust load enhanced in the both hemispheres but drastic change occurred in the southern hemisphere (reduction to enhancement). Other reason would be the difference of circumstances of the glaciogenic dust sites. In the southern hemisphere, glaciogenic dust is quickly transported over the Southern Ocean, where wind speed tends to be high and some of the dust could reach high

10  altitude. On the other hand, in the northern hemisphere, the glaciogenic dust emission occurs over the continents, where the glaciogenic dust is easier to deposit at the vicinity land grids and also the higher land fraction in the northern hemisphere possibly prevents dust to reach high troposphere. This is written in Sect. 3.1.

Technical comments

15  Abstract, line 15: remove 'found'.

It is removed.

Page 2, line 7: "Global warming is considered an important driver in investigations seeking to clarify the mechanisms of climate change, as stated

20  repeatedly by the Intergovernmental Panel on Climate Change (IPCC) in their assessment reports (IPCC, 2013)."
This doesn't really make sense. Perhaps change to:
"Global climate modelling is an essential tool in investigations seeking to clarify the mechanisms of climate change, as stated in Intergovernmental Panel on Climate Change (IPCC) assessment reports (IPCC, 2013)."

25  Thank you for the suggestion. The sentence is modified to "Climate modelling is an essential tool in investigations seeking to clarify the mechanisms of climate change, as stated in Intergovernmental Panel on Climate Change (IPCC) assessment reports (IPCC, 2013)."

Page 4, line 15: Make this sentence more direct. Change: "Previous studies that have investigated the effect of glaciogenic dust have not taken

into account the feedback of the dust to the atmosphere via the oceanic thermohaline circulation. " to highlight something like: previous studies have included a dynamic ocean in this context, so the impacts on ocean circulation globally are unknown.

Thank you for your suggestion. The sentence is changed to

5   "Previous studies have not included a dynamic ocean in this context, so the impacts on global ocean circulation are unknown"

Page 4, line 16: Move this sentence: "Moreover, Lambert et al. (2013) identified the possibility of polar amplification attributable to dust." to earlier where you discuss polar amplification, and explain what Lambert did in 1 sentence or so.

10   We agree that this is located unsuitable part of introduction. Now, A sentence "Lambert et al 2013 demonstrated two General Circulation Models coupled with online aerosol models underestimated dust flux and radiative forcing globally but especially pronounced over the polar regions and suggested the possibility of underestimate of polar amplification for LGM." Is inserted after the reviewing Mahowald and Takemura's studies.

Page 6, line 9: Change "and it was emitted constantly independent of the other conditions ", to "The fluxes estimated by Mahowald were added as
15   time-invariant sources into the simulations, and are not dependent on modelled land surface or atmospheric conditions".

Thank you for your suggestion. The sentence is modified as you suggested.

Page 12, line 19: "The panels clarify that the effect of snow ageing on the radiative perturbation is minor."
20   This sentence seems to contradict the results shown in figure 6 and the discussion starting on page 10 line 18?

Figure C shows that the radiative perturbation by dust loading in the atmosphere. Similarity of the right and left columns of Figure C suggests that the radiative perturbation by dust in the atmosphere is independent from ageing effect of snow surface. Hence, the surface warming in the northern hemisphere is not by dust aloft but by the deposited dust on snow. The sentence is modified to "The panels clarify that the effect of snow
25   ageing is independent from radiative perturbation by dust load in the atmosphere."

Page 14, line 15-16: You might mention that some of this model-data disagreement is related to reconstructed warming in Alaska, which is not

resolved in either LGM or LGMglac.

Thank you for the suggestion. A sentence is modified as follows, "Pronounced discrepancy is evident in the northern high latitudes around 70 °N with some proxy data over Alaska suggesting warmer temperatures than PI, which is not resolved in all our LGM experiment and the other LGM experiment of the PMIP3 models.".

Page 17, line 3: I don't think Claquin et al 2003, looked at snow albedo effect of dust?

Thank you for pointing it out. Their model includes ageing effect of snow following Douville et al. (1995a, b). However, they did not mention how the ageing affected specifically. We decide this is not a crucial sentence and thus removed it.

Page 17, line 5-7. To support this statement, you need to include the LGMglac.a – PI.a anomaly in figure 6.

Thank you for your suggestion. Figure 6 is expanded including LGM-PI panels and explained in Sec. 3.2.

**Response for the Anonymous Referee #2 on "The effect of high dust amount on the surface temperature during the Last Glacial Maximum: A modelling study using MIROC-ESM" by Rumi Ohgaito et al.**

We wish to express our appreciation for the referee for the constructive comments and suggestions. We modified the manuscript following your suggestions one by one.

The referee's comments are written in black and our replies are written in characters in blue color.

The revised version of the manuscript is clearly improved in comparing to existing work and organizing the discussion of some key variables. I still have a few minor comments dealing with the clarification of certain aspects discussed in the main text.

6/2-5: could you mention explicitly what those values are?

The refractive index of dust aerosol is set 1.530-2.00 x 10-3i at 0.55 micrometer dust. The number is added in the manuscript. The formulations concerning number concentration of cloud droplets and ice crystals are in Takemura et al. 2005, 2009. The references are inserted in the manuscript.

12/7: Please clarify that you refer to the surface, e.g. "The change at the surface is similar …"

"at the surface" is inserted following your suggestion.

12/3-8: You compare to other studies for surface values, but what at TOA? See also a recent review (Albani, S., Balkanski, Y., Mahowald, N. et al. Curr Clim Change Rep (2018) 4: 99. https://doi.org/10.1007/s40641-018-0100-7)

Thank you for letting us know about a latest review paper. Albani et al. (2018) is included in the introduction and this part of the text and discussion is included.

16/5: The net RADIATIVE effect … is negative

Thank you for your suggestion. The "radiative" is inserted.

16/16-17: This sentence is not very clear, please rephrase

5    The sentence is modified as the following.

"The effect of mineral dust aerosol on climate is highly uncertain but cooling is relatively likely (IPCC, 2013). Our results suggest the effect of dust on climate is dependent on background condition. However, our glaciogenic dust worked different from that demonstrated by Mahowald et al. (2006b) in the zonal mean."

10   17/9-10: An alternative explanation is that some of these observational estimates might affected by significant uncertainties, notably related to the potential the presence of non-aeolian material such as ice rafted debris (Kohfeld et al., 2013)

Kohfeld et al. 2013 said that they excluded marine sediment sites affected by IRD. However, we agree that it is worth mentioning it here.

15   18/1: I do not understand this sentence, please rephrase

Thank you for pointing it out. We agree that this sentence doesn't make sense. This and the next sentences are removed.

Figure 12: This relevant figure is quite hard to read. My suggestion would be to try to size it up and/or remove the line contours, which are largely
20   redundant with the color contours and makes it quite difficult to distinguish the circles corresponding to the observations

The figures are replaced with the version of no contours for (a) (b) (d) (e). Thank you for your suggestion.

Table 1: the integration length column is only indicated for panel (a). Pleas add to panel (b) as well.

The integration length for .a experiments are written in the text but in order to clarity, the column is added in Table 1 (b).

[revised manuscript text omitted]

---

## Author Response (AR3)

**Response for the Editor on "The effect of high dust amount on the surface temperature during the Last Glacial Maximum: A modelling study using MIROC-ESM" by Rumi Ohgaito et al.**

Editor's comments to the Author:

5  The new manuscript is largely improved and the comments from the reviewers have been thoroughly taken into account. The manuscript is therefore good for publications after the following minor corrections (page numbering from the authors's response to the reviewers, with manuscript starting on page 7).

We wish to express our appreciation for the editor for the careful comments and suggestions. We modified the manuscript following

10  your suggestions one by one.

The editor's comments are written in black and our replies are in characters in blue color.

Minor corrections.

1. Introduction

15  page 8, line 1: was => is

changed

page 9, line 2: remove comma after "i.e."

removed

page 10, lines 2-4: reformulate the sentence, for instance: Lambert et al. (2013) used two General Circulation Models coupled with online aerosol models and obtained underestimated dust flux and radiative forcing. This underestimation was global, but more

25  pronounced over the polar regions and they suggested the possibility that it contributes to an underestimation of polar amplification for LGM and future projections.

The sentence is replaced to what you suggested.

page 10, line 5: supposed => defined

changed

page 10, line 12: during => for (the LGM) and in => for (the PI)

changed as you suggested

page 11, line 2: addressed => addresses

changed

2. Model and experimental design

2.1: description of the MIROC-ESM

page 11 lines 13-14: "prognosis of" => computing

changed

page 12, line 7: insert commas before and after "as their mass mixing ratios"

inserted

page 12, line 18: replace "weighted to" by "weighted as a function of"

Changed, following your suggestion.

2.2 Experimental design

page 13, line 13: please insert Fig A of the supplementary material in the main text.

The sentence is modified to Fig. A in the main text.

3. Results

3.1 Dust amount and comparison with data archives

page 14 (and page 13, line 15 and 16). The tables are not listed in order

10   The numbering of the tables is changed to in order.

page 15, line 2: "but the load (38 Tg) is significantly smaller (62Tg)". Please clarify 38Thg is smaller than 62 Tg, and it is difficult to know which load corresponds to which simulation.

15   This part of the sentence is changed to more precisely to "but the load of LGMglac.a (39 Tg) is about 60 % of Mahowald's loading (62 Tg)".

3.2 Surface temperature at LGM and the effect of glaciogenic dust

page 16, line 5: latitude => latitudes

changed

page 16, line 5, insert "most" before "promnounced"

25   inserted

page 16, line 7: insert "results in" after "LGM" and replace "in consistent distribution to" by "and is consistent with the distribution of"

changed following your suggestions

page 16, line 14: I think the reference is to Figure 7g, not 7c

5  Thank you for pointing out the error. It is corrected.

page 17, line 6: please inset Fig B from supplementary material in main manuscript

The sentence is modified to Fig. B in the main text.

page 17, line 11: the reference to Krinner is missing; replace "on the point of" by "about"

Thank you for pointing out the error. The reference is added and the text is changed.

15  page 17, line 13: isopleth => isotherm

changed.

3.3 aerosol-radiation and aerosol-cloud interactions by dust
20  page 18, section title: would "dust-related aerosol-radiation and aerosol-cloud interactions" be better?

The section title is changed following your suggestion.

page 18, line 3: add "including dust impacts" after "standard experiment"?

added

page 18, line 5: clarification is needed here: is the aerosol-cloud interaction estimated with the comparison of the standard experiment with another experiment without dust at all? or without dust in the cloud radiation package?

The text is changed to "without dust at all".

page 18, line 10: insert "results from" before "previous studies"

inserted

10    page 19, line 16: insert "due to dust" after "perturbation"

inserted

3.4 influence of glaciogenic dust on the ocean

15    page 20, lines 16-17: the sentence "The different behaviour … LGMglac.e." can eb removed, as the idea is developed later in that section.

The sentence is deleted following your suggestion.

20    page 21, line 8: insert "for LGMglac.e compard to LGM.e" after "increased air temperature"

inserted

marked up manuscript follows:

[revised manuscript text omitted]